# Gasotransmitter modulation of hypoglossal motoneuron activity

**Brigitte M Browe**[1,2,3], **Ying-Jie Peng**[1,3], **Jayasri Nanduri**[1,3], **Nanduri R Prabhakar**[1,2,3], **Alfredo J Garcia III**[1,2,3]*

[1]Institute for Integrative Physiology, University of Chicago, Chicago, United States; [2]The University of Chicago Neuroscience Institute, The University of Chicago, Chicago, United States; [3]Department of Medicine, Section of Emergency Medicine at The University of Chicago, University of Chicago, United States

**Abstract** Obstructive sleep apnea (OSA) is characterized by sporadic collapse of the upper airway leading to periodic disruptions in breathing. Upper airway patency is governed by genioglossal nerve activity that originates from the hypoglossal motor nucleus. Mice with targeted deletion of the gene *Hmox2*, encoding the carbon monoxide (CO) producing enzyme, heme oxygenase-2 (HO-2), exhibit OSA, yet the contribution of central HO-2 dysregulation to the phenomenon is unknown. Using the rhythmic brainstem slice preparation that contains the preBötzinger complex (preBötC) and the hypoglossal nucleus, we tested the hypothesis that central HO-2 dysregulation weakens hypoglossal motoneuron output. Disrupting HO-2 activity increased the occurrence of subnetwork activity from the preBötC, which was associated with an increased irregularity of rhythmogenesis. These phenomena were also associated with the intermittent inability of the preBötC rhythm to drive output from the hypoglossal nucleus (i.e. transmission failures), and a reduction in the input-output relationship between the preBötC and the motor nucleus. HO-2 dysregulation reduced excitatory synaptic currents and intrinsic excitability in inspiratory hypoglossal neurons. Inhibiting activity of the CO-regulated $H_2S$ producing enzyme, cystathionine-γ-lyase (CSE), reduced transmission failures in HO-2 null brainstem slices, which also normalized excitatory synaptic currents and intrinsic excitability of hypoglossal motoneurons. These findings demonstrate a hitherto uncharacterized modulation of hypoglossal activity through mutual interaction of HO-2/CO and CSE/$H_2S$, and support the potential importance of centrally derived gasotransmitter activity in regulating upper airway control.

## Editor's evaluation

This is among the first papers to comprehensively describe a signaling pathway in motor neurons and the consequences of its deficiency. The finding of gaseous neurotransmitter controlling motor neuron function is novel and opens the door for investigations into other motor neuron pools.

## Introduction

Sporadic airway collapse is a hallmark of obstructive sleep apnea (OSA), a prevalent breathing disorder estimated to affect nearly a billion people throughout the world *Lyons et al., 2020*; *Malhotra et al., 2021*. When left untreated, OSA predisposes the individual to a variety of diseases including hypertension *Mehra, 2019*; *Yeghiazarians et al., 2021*, diabetes *Loffler et al., 2020*; *Hua, 2020*, and cognitive decline *Daulatzai, 2017*; *Bibbins-Domingo et al., 2017*. Multiple factors contribute to the genesis of OSA including compromised pharyngeal anatomy *Castro and Freeman, 2021*; *Genta et al., 2017*, inadequate upper airway muscle function *Neelapu et al., 2017*; *Vos et al., 2010*; *Kubin,*

*For correspondence:
ajgarcia3@uchicago.edu

Competing interest: The authors declare that no competing interests exist.

*2016*, low arousal threshold *Eckert et al., 2013*, and a hypersensitive chemoreflex (i.e. high loop gain) *Nemati et al., 2011*.

Peng et al. recently reported that mice with deletion of the *Hmox2* gene, which encodes the enzyme heme oxygenase 2 (HO-2), exhibit a high incidence of OSA *Peng et al., 2017*. OSA in HO-2 null mice was attributed, in part, to increased loop gain arising from the heightened carotid body chemoreflex *Peng et al., 2017*; *Peng et al., 2018*; *Osman et al., 2018*; *Prabhakar and Semenza, 2012*. While HO-2 activity produces several bioactive molecules *Mancuso, 2004*, the loss of HO-2 dependent carbon monoxide (CO) production was shown to be a primary driver of the enhanced carotid body chemoreflex and the subsequent OSA phenotype *Peng et al., 2017*. However, output from the hypoglossal motor pool can be influenced by multiple factors, including HO-2-mediated action from within the central nervous system itself.

Loss of neuromuscular control over upper airway muscles has a key role in producing obstructive apneas *Schwartz et al., 2008*; *Horner, 2001*. Disrupting neuronal excitability in the hypoglossal nucleus that is responsible for genioglossal nerve activity increases the likelihood for the tongue to occlude the upper airway during inspiration. Such disruptions may involve changing the state-dependent balance between excitation and inhibition received by hypoglossal motoneurons *Horner, 2009* and/or by directly modulating their intrinsic excitability *Horton et al., 2017*; *Fleury Curado et al., 2017*; *Fleury Curado et al., 2018*. It is, however, unknown how HO-2 signaling at the level of the preBötC and the hypoglossal nucleus influence these factors contributing to upper airway tone and patency.

We tested the role of central HO-2 signaling in influencing hypoglossal motor output using a combination of electrophysiological, genetic, and pharmacological approaches in rhythmic medullary brainstem slice preparations. Dysregulated HO-2 activity increased the occurrence of subnetwork activity in the preBötC, which was associated with an increased cycle-to-cycle irregularity of rhythmogenesis. In hypoglossal motoneurons, excitatory synaptic drive currents and intrinsic excitability were reduced by HO-2 dysregulation. These phenomena also coincided with circuit level effects. HO-2 dysregulation diminished the input-output relationship and increased the likelihood of transmission failure between the preBötC activity and the hypoglossal nucleus. These effects of HO-2 dysregulation could be mimicked by exogenous $H_2S$ and mitigated by either pharmacological inhibition or genetic ablation of CSE. Together these observations indicate that centrally derived HO-2/CO and CSE/$H_2S$ signaling interact as important modulators of hypoglossal output contributing to upper airway tone.

## Results

### Hypoglossal neurons express hemeoxygenase-2 (HO-2)

We assessed whether hypoglossal neurons express HO-2. Hypoglossal neurons showed positive immunohistochemical expression for HO-2 as indicated by co-localization of HO-2 with ChAT, a motoneuron marker (*Figure 1A*, n=3).

### Disrupting HO-2 function impairs hypoglossal inspiratory activity

Two approaches were employed to assess the role of HO-2. First, using Cr(III) Mesoporphyrin IX chloride (ChRMP459, 20 μM) as a pharmacological inhibitor of HO *Ding et al., 2008* in wild-type slices and second, with a genetic approach using brain slices from HO-2 null mice. Simultaneous extracellular field recordings were performed from preBötC and the hypoglossal nucleus in wild-type slices prior to and during ChRMP459 exposure (n=34 slices). ChRMP459 promoted the emergence of subnetwork activity (i.e., integrated bursts ≤ 50% of mean integrated burst amplitude during baseline; demarcated by hash symbol, *Figure 1B*) in the preBötC. This subnetwork preBötC activity commonly failed to drive corresponding output hypoglossal nucleus (highlighted in pink, *Figure 1B*) and was associated with an increase in the irregularity score of amplitude ($IrS_{AMP}$) in the preBötC (*Figure 1C*, *bottom*; Baseline: 0.279±0.39, ChRMP459: 0.406±0.05, p=0.021). However, ChRMP459 did not impact the $IrS_{AMP}$ in hypoglossal nucleus (*Figure 1C*, *top*; Baseline: 0.385±0.05, ChRMP459: 0.372±0.03, p=0.701) nor affect the frequency or amplitude of the integrated bursts from either brainstem network (*Table 1*).

ChRMP459 caused a consistent reduction in the cycle-to-cycle input-output relationship between preBötC and the hypoglossal nucleus (*Figure 1D*) which was quantified by comparing input-output (I/O) ratios prior to and during ChRMP459 (Baseline: 1.06±0.04 vs. ChRMP459: 0.592±0.06; p<0.0001).

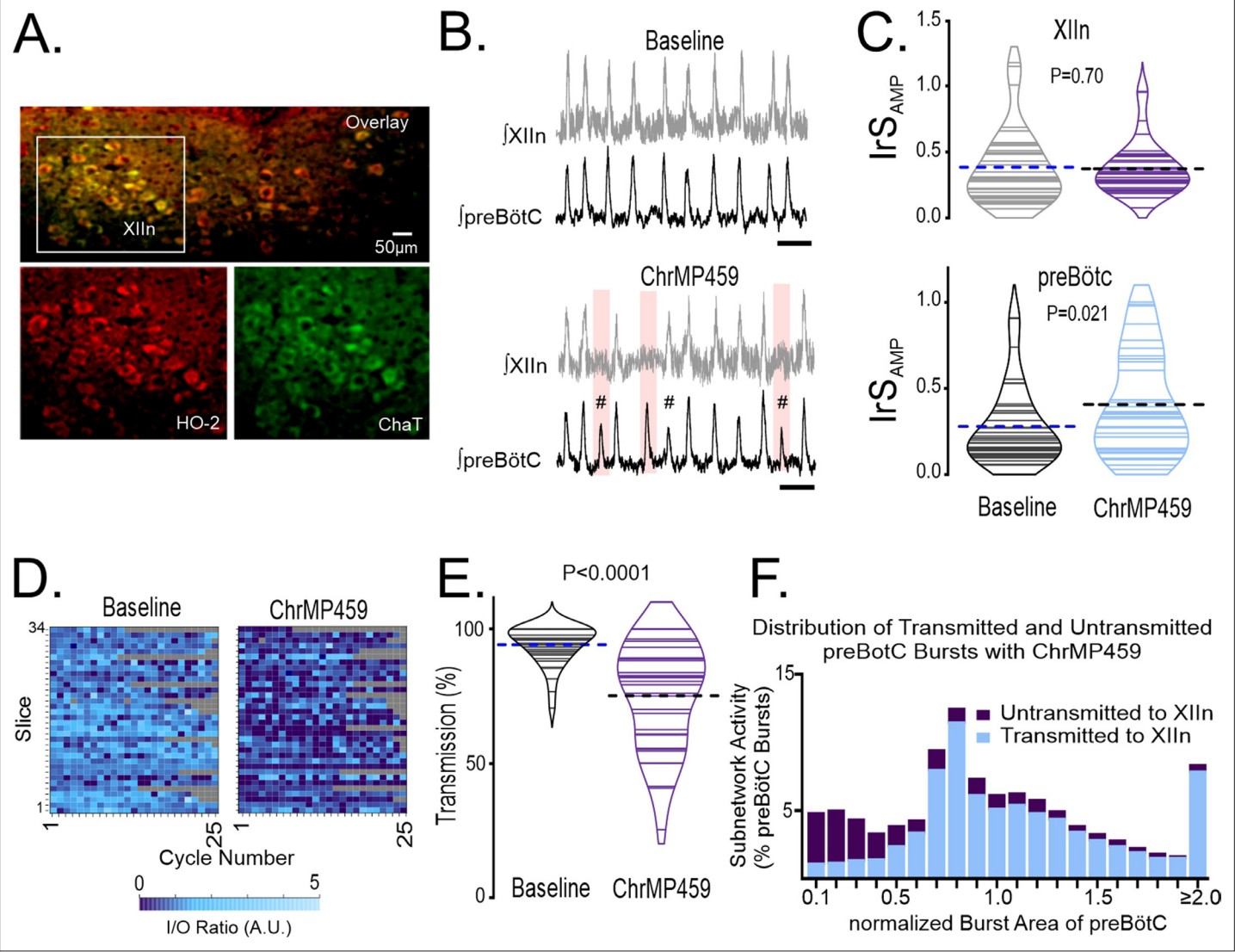

**Figure 1.** Disruption of hemeoxygenase-2 (HO-2) impairs inspiratory activity from the hypoglossal nucleus and the preBötC. (A) HO-2 (red bottom left) expression co-localized to ChAT[+] cells (green bottom right) of the hypoglossal nucleus (XIIn, overlay top, n=3). (**B–F**) Population recordings of rhythmic brain slices were made from ipsilateral preBötC and XIIn simultaneously, analyses were performed in baseline and during bath application of 20 μM ChrMP459. (**B**) Integrated traces of network activity in spontaneously rhythmic brainstem slices (n=34) recorded from XIIn (gray) and preBötC (black) before (top) and during (bottom) ChrMP459. Failed transmission events are highlighted (pink box) and subnetwork preBötC activity (#, defined as preBötC events with an integrated burst area ≤50% of the mean integrated burst area in baseline and are #) are evident in ChrMP459. Scale bar: 5 s. (**C**) Comparison of integrated amplitude irregularity score (IrS$_{AMP}$) during baseline and in ChrMP459 from the XIIn (top) and the preBötC (bottom, n=34). Solid lines within violin plots illustrate IrS$_{AMP}$ from individual experiments. Thick dashed line illustrates mean IrS$_{AMP}$. (**D**) Heat maps of I/O ratios from 25 consecutive cycles in baseline and ChrMP459. Each row reflects an individual experiment while each cell represents the I/O ratio for a given cycle. Gray boxes indicate non-events from recordings from slower rhythms where less than 25 cycles occurred during the analysis window. (**E**) Comparison of transmission from preBötC to XIIn between Baseline (gray) and ChrMP459 (purple). Solid lines within violin plots illustrate transmission from individual slices. Thick dashed line illustrates mean transmission value. (**F**) Distribution of transmitted (blue) and untransmitted (purple) preBötC bursts in ChrMP459. These values are expressed as a percentage of the total of preBötC events detected from all experiments and are binned. Bin interval = 0.1 intervals of the normalized integrated burst area. preBötC bursts were normalization to the mean integrated burst area during baseline. Statistical analysis for all comparisons via paired t-test; error bars: Standard Error Measurement (SEM); significance level P<0.05.

A reduction of input-output relationship between preBötC and the hypoglossal nucleus has been previously associated with increased transmission failure, which is defined by the inability of preBötC activity to produce hypoglossal output at the network level *Garcia et al., 2016*. The reduced I/O ratio in ChrMP459 was, indeed, associated with increased transmission failure (*Figure 1E*, Baseline: 94.130±1.27% vs. ChrMP459: 75.100 ± 3.43%, p<0.0001). Examining the relationship between failed

**Table 1.** Properties of Network Activity in the preBötC and XIIn during dysregulated HO-2.

Analysis of instantaneous frequency ($f_{inst}$) and integrated burst amplitude of network activity in the preBötC and hypoglossal nucleus. Statistical analysis for all comparisons via paired t-test. Values are displayed as mean ± SEM (n). Significance level P<0.05.

| Experiment | Recording Location | $f_{inst}$ (Hz) | | | Burst Amplitude (mV) | | |
|---|---|---|---|---|---|---|---|
| | | Control* (n) | Dysregulated HO-2 (n) | p-value | Control* (n) | Dysregulated HO-2 (n) | p-value |
| ChrMP459 | preBötC | 0.225±0.014 (34) | 0.242±0.019 (34) | 0.139 | 0.078±0.011 (34) | 0.08±0.012 (34) | 0.626 |
| HO-2 null slice | preBötC | 0.235±0.022 (18) | 0.420±0.049 (11) | 0.0006 | 0.059±0.007 (18) | 0.051±0.010 (11) | 0.463 |
| ChrMP459 | XIIn | 0.239±0.015 (34) | 0.221±0.015 (34) | 0.044 | 0.047±0.008 (34) | 0.046±0.009 (34) | 0.804 |
| HO-2 null slice | XIIn | 0.256±0.018 (18) | 0.432±0.051 (11) | 0.0006 | 0.038±0.006 (18) | 0.021±0.003 (11) | 0.050 |

*= Control in ChrMP459 experiments is defined as baseline activity prior to ChrMP459. Control in HO-2 null experiments is defined as recordings in wild-type slices.

transmission and the burst area in the preBötC revealed that while the many transmission failures were associated with subnetwork preBötC activity (i.e. integrated burst areas ≤ 50%), these events were not restricted to subnetwork activity but rather, occurred across a range of integrated burst areas from the inspiratory network (*Figure 1F*). Thus, HO inhibition appeared to produce a generalized weakening in the relationship between preBötC and hypoglossal nucleus activity evident across the spectrum of different burst areas generated in the preBötC.

As ChrMP459 cannot distinguish activities between heme oxygenase isoforms, we compared rhythmic activities in brain slices from wild-type mice (n=18) and HO-2 null mice (n=11) to assess the contribution of HO-2. Both the intermittent occurrence of subnetwork activity in the preBötC (demarcated by hash symbol, *Figure 2A*) and failed transmission events were observed in HO-2 null slices (highlighted in pink, *Figure 2A*). preBötC activity in HO-2 null mice was associated with an increased IrS$_{AMP}$ in both the preBötC (*Figure 2B*, WT: 0.323±0.02, HO-2 null: 0.419±0.03, p+0.014) and the hypoglossal nucleus (*Figure 2C*, WT: 0.269±0.03, HO-2 null: 0.534±0.12, p=0.018). While the frequency was faster in HO-2 null rhythms relative to wild-type rhythms, the burst amplitude was not different in either preBötC or the hypoglossal (*Table 1*).

Relative to wild-type slices, the HO-2 null preparations had smaller cycle-to-cycle I/O ratios (*Figure 2D*; wild-type: 1.023±0.03 vs. HO-2 null: 0.799±0.07, p=0.002); this was accompanied by a smaller percentage of transmission in HO-2 null slices (*Figure 2E*, wild-type: 98.210±0.87% vs. HO-2 null: 64.110 ± 6.00%, p<0.0001). Similar to ChrMP459 experiments, while subnetwork activity led to many failed transmission events, failed transmission occurred across a range of burst areas from the preBötC (*Figure 2F*). These findings were consistent with ChrMP459 findings and illustrated that lost HO-2 activity is sufficient for promoting subnetwork preBötC activity, reducing input-output relationship between preBötC and the hypoglossal nucleus, and increasing transmission failures. Given these similarities and the limited availability of HO-2 null mice, several of the following studies were performed using the ChrMP459 in rhythmic wild-type brainstem slices.

## HO inhibition does not affect intermediate premotor neuron activity

Intermediate premotor neurons relay drive from the preBötC to the hypoglossal nucleus *Koizumi et al., 2013*; *Revill et al., 2015*. Therefore, it was possible that HO inhibition impaired transmission of drive from the preBötC by perturbing activity from intermediate premotor neurons. To address this possibility, simultaneous extracellular recordings (n=5) were made from the preBötC, the field of the ipsilateral premotor neurons, and the hypoglossal nucleus (*Figure 3A*, *left panel*). Baseline transmission from the preBötC to the premotor field and to the hypoglossal nucleus was reliable and consistent (*Figure 3A*, *middle panel*). In ChrMP459, intermittent failures of hypoglossal nucleus activity corresponding preBötC activity were evident yet during these failed cycles preBötC activity still produced detectable network activity in the field of intermediary premotor neurons (*Figure 3A*, *right panel*). Indeed, while neither the cycle-to-cycle I/O ratio nor transmission from the preBötC to the premotor field was affected by ChrMP459 (*Figure 3B*: *left*; I/O ratio: Baseline: 1.116±0.09 vs ChrMP 1.162±0.15, p=0.816; *right*; Transmission: Baseline: 100.0±0.0% vs ChrMP 86.350 ± 11.81%, p=0.312), the HO inhibitor reduced the cycle-to-cycle I/O ratio and the transmission between the

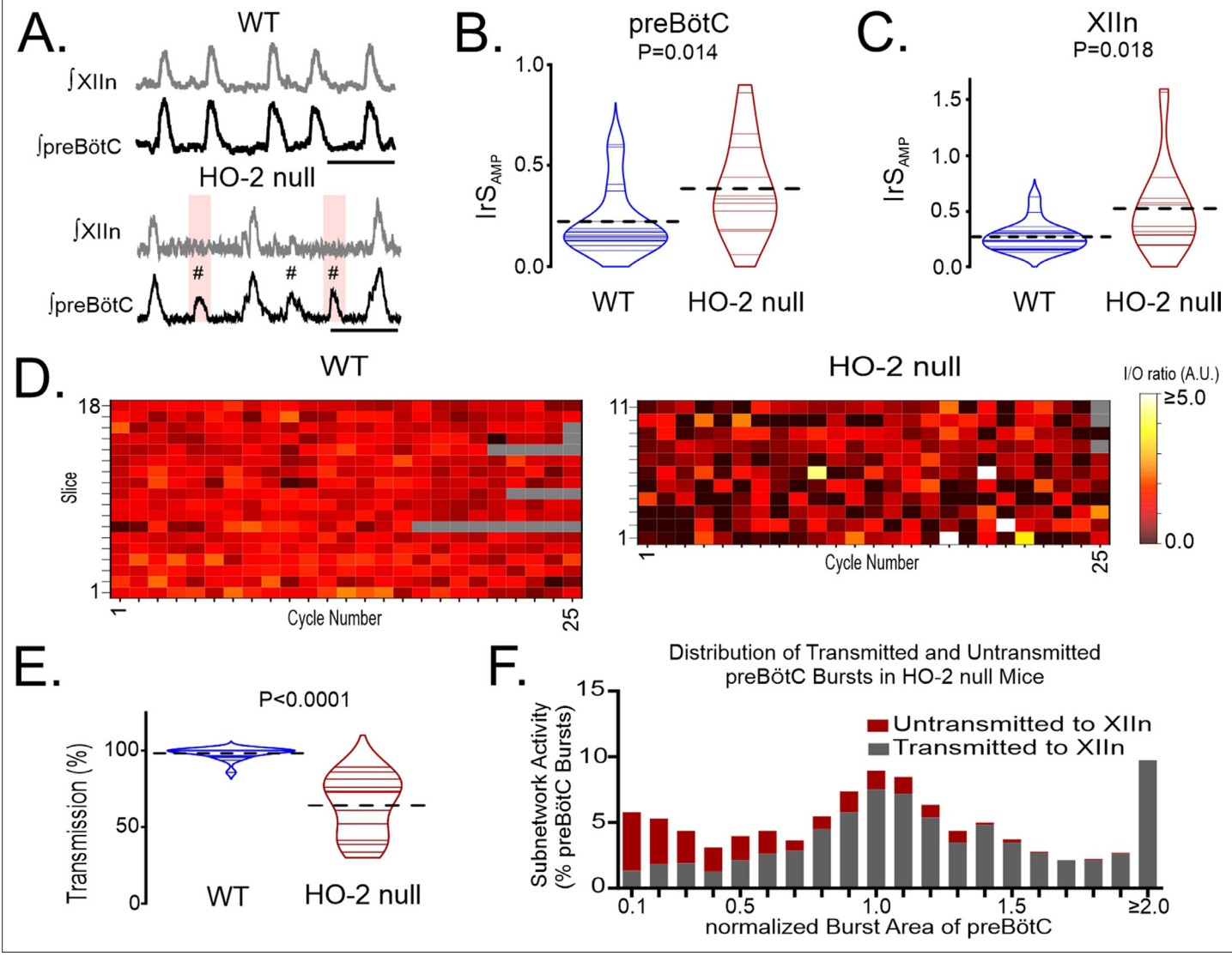

**Figure 2.** Genetic deletion of HO-2 reduces the I/O relationship between preBötC and the hypoglossal nucleus and uncouples of motor output from inspiratory rhythmogenesis. (**A**) Representative integrated traces of network rhythms in the preBötC and XIIn from wild-type (WT; left, n=18) and HO-2 null (right, n=11) slices. Failed transmissions (pink box) and subnetwork preBötC activity (#) are evident in HO-2 null slices. Scale bar: 4 s. (**B**) Comparison of $IrS_{AMP}$ in the preBötC of WT (blue) and HO-2 null (red) slices. (**C**) Comparison of $IrS_{AMP}$ in the XIIn of WT (blue) and HO-2 null (red) slices. (**D**) Heat maps of cycle-to-cycle I/O ratios from individual experiments performed in WT (left) and HO-2 null (right) slices. Gray boxes indicate non-events in recordings from slower rhythms where less than 25 events occurred during the analysis window. (**E**) Comparison of transmission from preBötC to XIIn between WT (blue) and HO-2 null (red) slices. Thin gray and purple lines illustrate transmission from individual slices. Thick dashed lines illustrate mean transmission value. (**F**) Distribution of transmitted (gray) and untransmitted (red) preBötC bursts in HO-2 null slices. These values are expressed as a percentage of the total of preBötC events detected from all experiments and are binned. Bin interval = 0.1 intervals of the normalized integrated burst area. preBötC bursts were normalization to the mean integrated burst area from each individual recording. Statistical analysis for all comparisons via unpaired t-test; error bars: SEM; significance level P<0.05.

premotor field and the hypoglossal nucleus (*Figure 3C*: *left*, I/O: Baseline: 1.102±0.18 vs ChrMP 0.401±0.09, p=0.041; *right,* Transmission: Baseline 89.570±4.34% vs ChrMP 57.620 ± 13.69%, p=0.041). Thus, these findings suggested that ChrMP459 potentially affected synaptic properties to the hypoglossal motoneurons and/or the intrinsic excitability of hypoglossal neurons.

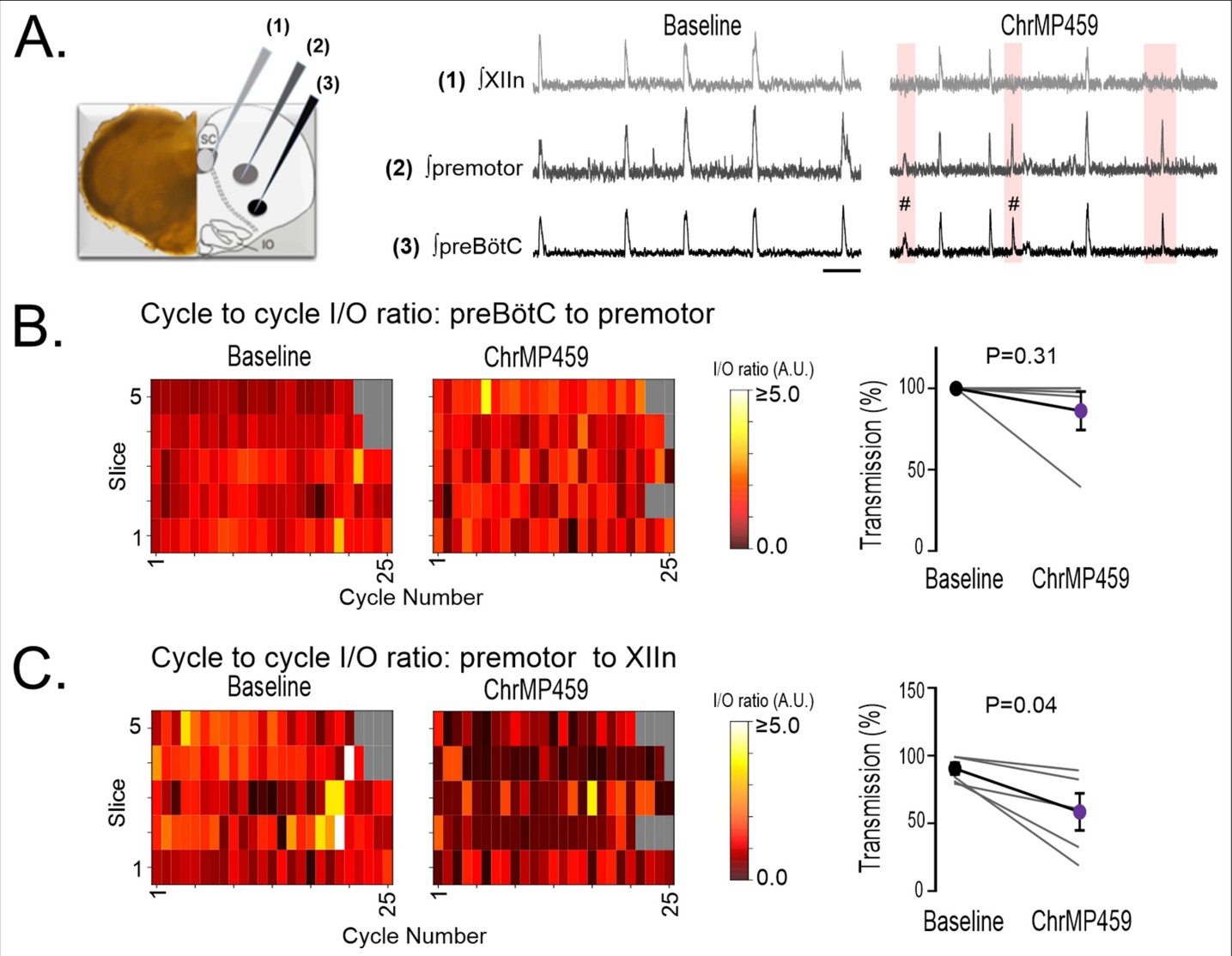

**Figure 3.** While ChrMP459 does not change transmission from the preBötC to the premotor area, ChrMP459 increases transmission failure from the premotor area to the hypoglossal nucleus. (**A**) Diagram of medullary brain slice illustrating relative electrode placement for simultaneous triple extracellular recordings (n=5) from the XIIn (light gray, 1), premotor field (dark gray, 2), and preBötC (black, 3). Corresponding representative traces of integrated network activity in Baseline (left) and in 20 μM ChrMP459 (right). Failed transmission (pink box) from preBötC to XIIn and preBötC subnetwork burst activity (#) are evident in ChrMP459; scale bar: 5 s. (**B**) Heat maps of the cycle to cycle I/O ratio from individual slices (left) and transmission (right) between preBötC and the premotor field. (**C**) Heat maps of the cycle to cycle I/O ratio from individual slices (left) and transmission (right) between the premotor field and XIIn. Statistical analysis for all comparisons via paired t-test; error bars: SEM; significance level P<0.05.

## HO inhibition suppresses inspiratory drive currents and reduces excitability in hypoglossal neurons

To assess the effect of HO inhibition on postsynaptic activity of the hypoglossal neurons, we performed patch clamp recordings from a total of 27 wild-type hypoglossal neurons exposed to ChrMP459. These hypoglossal neurons were disinhibited from fast inhibition using picrotoxin (50 μM) and strychnine (1 μM), which allowed us to focus on inspiratory-related fast glutamatergic drive. Of the 27 hypoglossal neurons, 19 received excitatory synaptic drive in-phase with the preBötC (i.e. inspiratory hypoglossal neurons). Peak inspiratory drive currents were reduced in ChrMP459 (*Figure 4A*, n=19, Baseline: −142.90±22.82 pA vs. ChrMP459: −95.31±21.79 pA, p=0.004). Reduced drive coincided with hypoglossal neurons generating fewer action potentials per preBötC burst in ChrMP459 (*Figure 4B*, n=17, Baseline: 14.68±2.24 action potentials per burst vs. ChrMP459: 6.798±1.55 action potentials

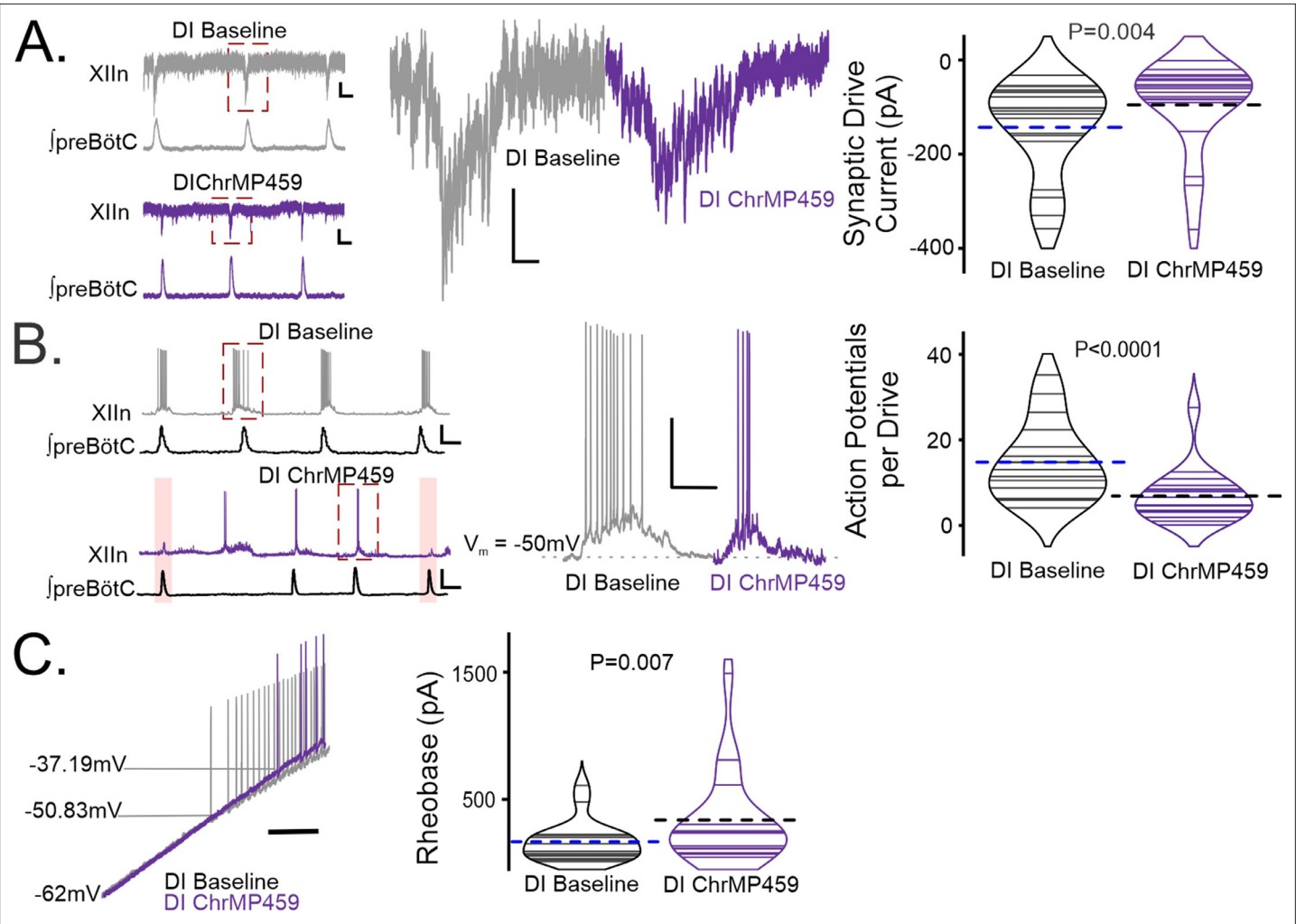

**Figure 4.** Heme oxygenase inhibition reduces inspiratory drive currents in hypoglossal neurons. Whole cell patch clamp recordings were made from hypoglossal neurons in rhythmic brain slices while simultaneously recording ipsilateral preBötC activity in Baseline and in ChrMP459. Neurons were disinhibited from fast synaptic inhibition using 50 µM PTX and 1 µM Strychnine (DI). (**A**) (left) Representative voltage clamp recordings from a XIIn neuron ($V_{holding}$ = –60 mV) aligned with corresponding integrated network activity from preBötC before (DI Baseline, top, gray) and after 20 µM ChrMP459 (DI ChrMP459, bottom, purple). Scale bar: 1 s x 10 pA. (middle) Magnification of highlighted (red dotted box) drive currents from DI Baseline (gray) and DI ChrMP459 (purple). Scale bar: 100 ms x 20 pA. (right). Comparison of XIIn inspiratory drive current magnitude distribution in DI Baseline (gray) and DI ChrMP459 (n=19, purple). Thin solid lines illustrate individual neuron response. Dashed black line illustrates mean drive current. (**B**) (left) Representative current clamp recordings from a spontaneously active XIIn neuron with the preBötC network rhythm in DI Baseline (top, gray) and DI ChrMP459 (bottom, purple); skipped transmission of action potentials in DI ChrMP459 are highlighted (pink box). Scale bar 2 s x 20 mV. (middle) Magnification of highlighted neuronal activity (red dashed box in trace, left). Scale bars: 100 msec x 25 mV. (right) Distribution of the average number of action potentials generated per inspiratory burst in DI Baseline (gray) and in DI ChrMP459 (n=17, purple). Thin solid lines illustrate individual neuron response. Dashed black line illustrates mean action potentials per drive. (**C**) (left) Representative trace of current clamp recording in response to ramp current injection during DI Baseline (gray trace) and in DI ChrMP459 (purple trace); scale bar: 500 ms. (right) Comparison of rheobase distributions found in inspiratory XIIn neurons during DI Baseline (gray) and in DI ChrMP459 (n=19, purple). Thin solid lines illustrate individual neuron response. Dashed black line illustrates mean Rheobase. Statistical analysis for all comparisons via paired t-test; error bars: SEM; significance level P<0.05.

The online version of this article includes the following figure supplement(s) for figure 4:

**Figure supplement 1.** ChrMP459 decreases rheobase of non-inspiratory hypoglossal neurons.

per burst, p<0.0001). Injection of a depolarizing ramp current into hypoglossal neurons revealed that the HO inhibitor increased rheobase among inspiratory hypoglossal neurons (***Figure 4C***, n=19, Baseline: 167.5±35.85 pA vs. ChrMP459: 338.0±82.50 pA, p=0.007) yet decreased rheobase in non-inspiratory hypoglossal neurons (i.e. neurons not receiving drive during preBötC activity; ***Figure 4—figure supplement 1***, n=8, Baseline: 280.5±56.43 pA vs. ChrMP459: 228.2±47.96 pA, p=0.0117).

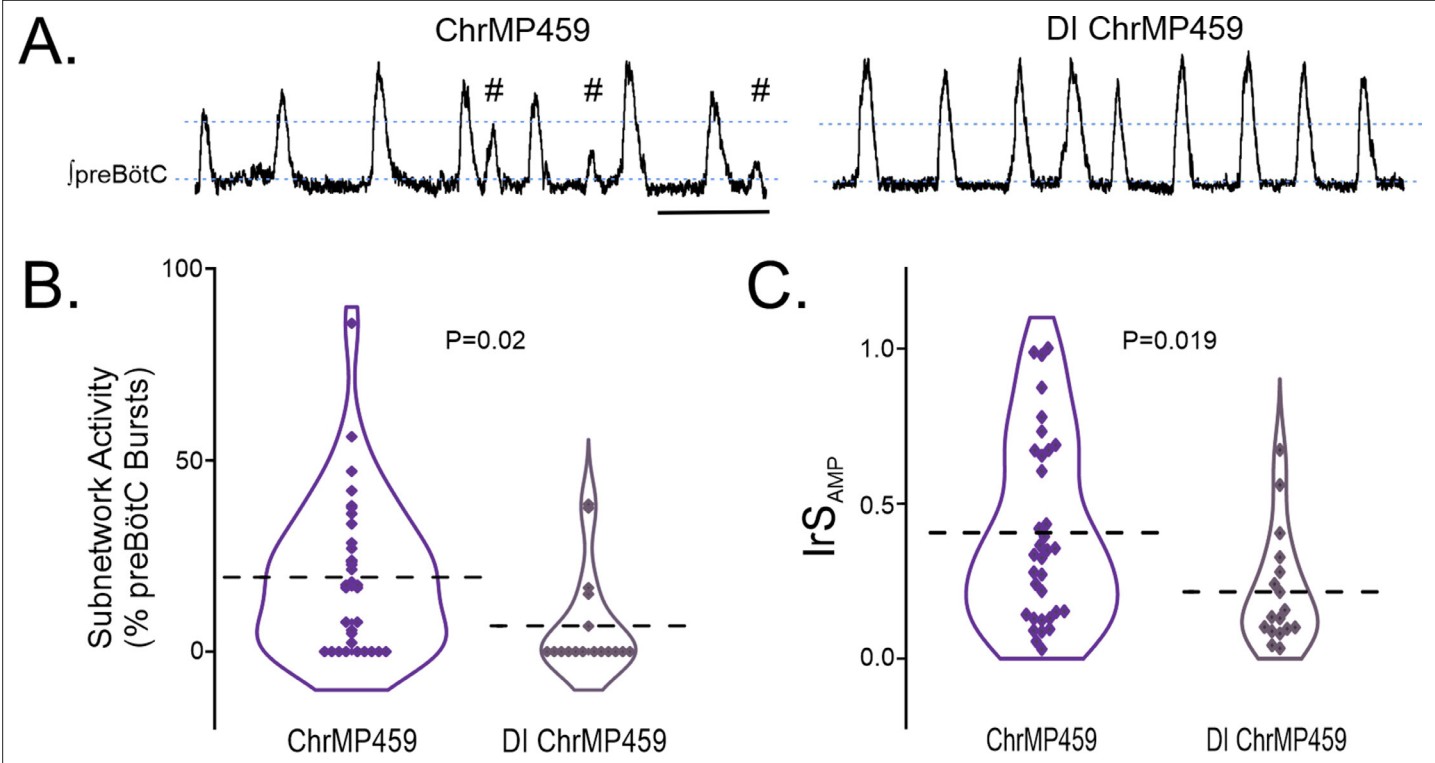

**Figure 5.** Disinhibition reduces ChrMP459-induced subnetwork activity and amplitude irregularities in preBötC population activity. (**A**) Representative integrated trace of preBötC activity from slices with bath application of ChrMP459 (top, n=34) and disinhibited slices with ChrMP459 and 50 µM PTX and 1 µM Strychnine for fast synaptic inhibition (DI ChrMP459, bottom, n=17). Subnetwork preBötC activity (#) is identified in ChrMP459. Scale bar 5 sec. (**B**) Comparison of subnetwork activity in preBötC slices as a percentage of total bursts (subnetwork ≥%50 average burst area in baseline) in ChrMP459 (purple) and DI ChrMP459 (dark purple). Individual values represented by ◊, thick dashed line represents mean subnetwork activity. (**C**) Comparison of IrS$_{AMP}$ between ChrMP459 (purple, replotted from *Figure 1C* preBötC) and DI ChrMP459 (dark purple). Individual values represented by ◊, thick dashed line represents mean IrS$_{AMP}$. Statistical analysis for all comparisons via paired t-test; error bars: SEM; significance level P<0.05.

## Disinhibition reduces occurrence of subnetwork activity and amplitude irregularities in the preBötC caused by HO inhibition

Examining the preBötC rhythm under disinhibited conditions also revealed that ChrMP459 appeared to cause fewer subnetwork bursts in the disinhibited preBötC (*Figure 5A*). Indeed, the percentage of subnetwork preBötC activity was greater in ChrMP459 when synaptic inhibition was preserved (*Figure 5B*, ChrMP459, n=34: 19.43±3.37% vs disinhibited ChrMP459, n=17: 6.72 ± 3.13%, p=0.02). Similarly, the IrS$_{AMP}$ during ChrMP459 was smaller in the disinhibited preBötC rhythm (*Figure 5C*, ChrMP459: 0.406±0.051 vs disinhibited ChrMP459: 0.215±0.04, p=0.019).

## Elevated levels of H$_2$S are observed in the hypoglossal nucleus of HO-2 null mice

We next sought to determine the mechanism(s) by which inhibition of HO-2 affected hypoglossal neuron activity. Earlier studies *Prabhakar, 2012*; *Morikawa et al., 2012* have reported that HO-2 is a negative regulator of CSE-dependent H$_2$S production. To test this possibility, we first examined whether the hypoglossal neurons express CSE. In the wild-type hypoglossal nucleus, CSE is expressed in ChAT-positive hypoglossal neurons (*Figure 6A*, n=3 mice). Homogenates made from hypoglossal and control tissue punches were prepared from wild-type and HO-2 mice to determine CSE-dependent H$_2$S abundance. Relative to the wild-type hypoglossal nucleus (*Figure 6B blue*; n=6, 60.58±6.37 nmol • mg$^{-1}$ • h$^{-1}$), H$_2$S abundance was greater in the hypoglossal nucleus of HO-2 null mice (*Figure 6B red*; n=6, 144.12±8.29 nmol • mg$^{-1}$ • h$^{-1}$, p<0.001), but not different from the inferior olive brainstem region of HO-2 null mice (*Figure 6B gray*; n=4, 56.10±2.88 nmol • mg$^{-1}$ • h$^{-1}$, p>0.05).

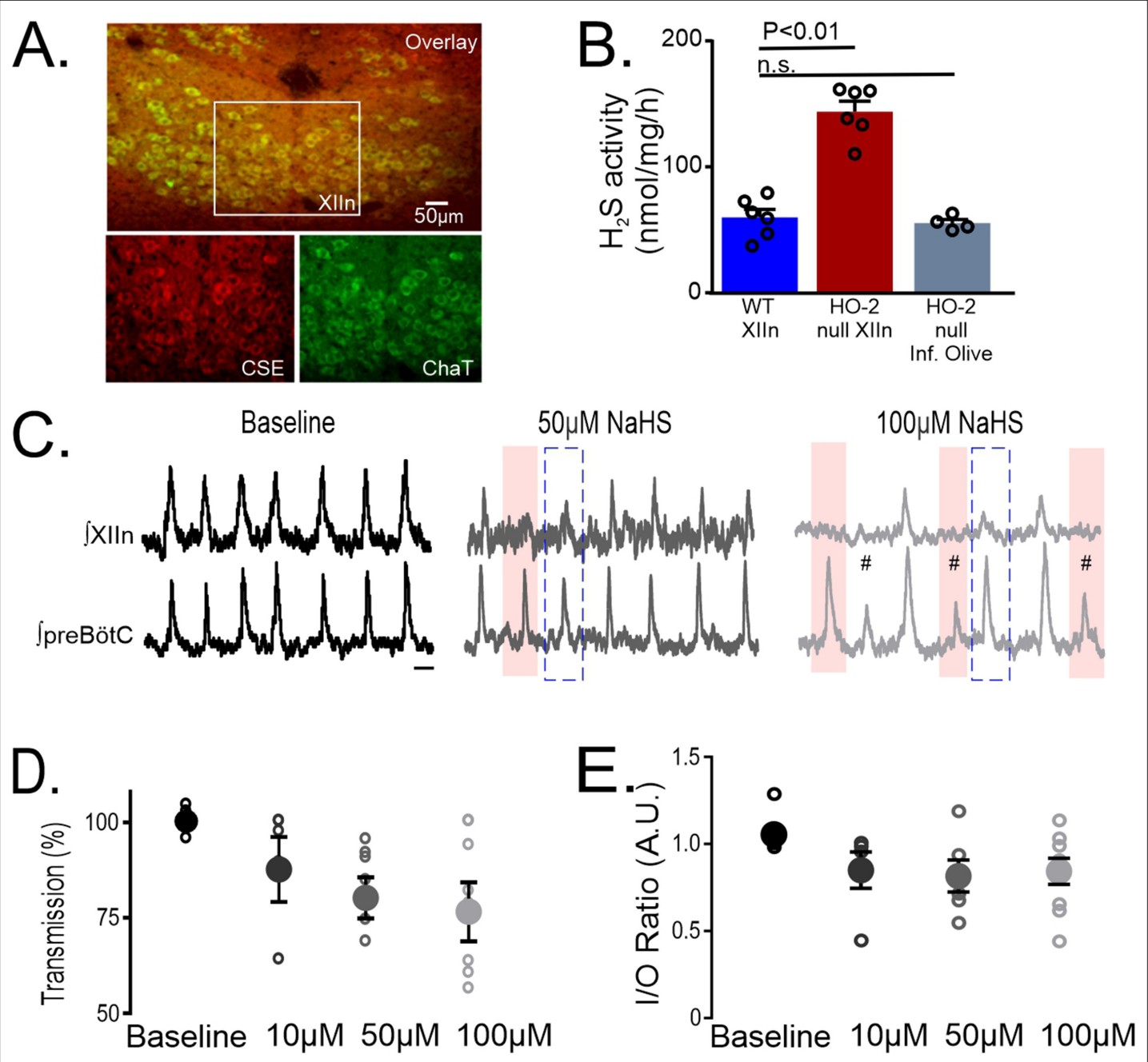

**Figure 6.** CSE-dependent $H_2S$ is produced in the hypoglossal nucleus and exogenous NaHS application uncouples hypoglossal nucleus activity from the preBötC. (**A**) CSE (red, bottom left) expression co-localizes with ChAT+ neurons (green, bottom right) in the XIIn (overlay, top). Scale bar 50 μm. (**B**) CSE-dependent $H_2S$ generation in pooled homogenates from WT and HO-2 null. Homogenates were prepared from tissue punches from the XIIn (red area in slice diagram) and inferior olive (gray area in slice diagram) at bregma between –7.20 mm and –7.90 mm. (WT: XIIn n=6; HO-2 null: XIIn n=6, inferior olive n=4). Each n in B represents a biological replicate consisting of the corresponding anatomical region pooled from two animals. (**C**) Integrated traces from XIIn (top) and preBötC (bottom) during Baseline (black), and in response to the $H_2S$ donor, NaHS, at 50 μM (dark gray) and 100 μM (light gray). NaHS application caused XIIn but not preBötC burst amplitude to diminish (blue dashed box) and in some cases, preBötC drive failed to produce activity in the XIIn (pink boxes). (**D**) Comparison of transmission from preBötC to XIIn after NaHS application at 10 μM, 50 μM and 100 μM. (**E**) I/O ratios for each NaHS concentration. (Baseline: n=9; 10 μM n=5; 50 μM n=6; 100 μM n=9). Statistical analysis for all comparisons via one-way ANOVA with Dunnett's correction; error bars: SEM; significance level P<0.05.

These findings show that the hypoglossal nucleus expresses CSE and suggests that HO-2 negatively regulates H$_2$S production in the hypoglossal nucleus.

## H$_2$S mediates impaired transmission of inspiratory drive caused by disrupted HO-2 function

If the impaired transmission of inspiratory drive to the hypoglossal nucleus by HO-2 dysregulation involves CSE-derived H$_2$S then: (1) an H$_2$S donor should mimic the effects of disrupted HO-2 activity; (2) CO administration should improve the input-output relationship in respiratory slices from HO-2 null mice and ChrM459 application; and, (3) CSE blockade should restore the transmission from the preBötC to the hypoglossal nucleus. The following experiments tested these possibilities.

Wild-type brainstem slices exhibited a nearly 1:1 ratio of preBötC activity to hypoglossal nucleus output (*Figure 6C*, *left*). Application of NaHS, a H$_2$S donor reduced this transmission from preBötC to hypoglossal output (*Figure 6C*, *middle, right*) in a dose-dependent manner (*Figure 6D*; 0 µM NaHS: n=9, 100.0 ± 0.77%; 10 µM NaHS: n=5, 90.14 ± 6.78%; 50 µM NaHS: n=6, 84.18 ± 4.29%; 100 µM NaHS: n=6, 81.26 ± 6.19%), which coincided with a reduction in I/O ratio by NaHS (*Figure 6E*; 0 µM NaHS: 1.055±0.03; 10 µM NaHS: 0.850±0.10; 50 µM NaHS: 0.816±0.09; 100 µM NaHS: 0.843±0.07). These findings demonstrated that increasing H$_2$S abundance reduces hypoglossal neuronal activity consistent with findings from experiments using ChrM459 or HO-2 null mice.

HO-2 dependent CO is known to inhibit CSE-dependent H$_2$S production *Prabhakar, 2012*; *Morikawa et al., 2012*. Therefore, we sought to assess how the pharmacological CO donor, CORM-3 (20 µM), impacted activity in ChrMP459-treated wild type rhythmic slices (*Figure 7A*, n=4) and rhythmic slices from HO-2 null mice (n=4). Dysregulated HO-2 activity, caused by either pharmacological (ChrMP459) or genetic (HO-2 null mice) manipulation, is improved by CORM-3 as indicated by augmented I/O ratios (*Figure 7B*, n=8; dysregulated HO-2: 0.705±0.09 vs. CORM-3: 1.05±0.07, p=0.01) and improved transmission (*Figure 7C*, dysregulated HO-2: 75.05±6.30% vs. CORM-3: 94.21 ± 2.67%, p=0.020).

To determine the involvement of CSE, inspiratory activity in the brainstem slice from HO-2:CSE double null mice appeared to be stable and consistent (*Figure 7D*). Quantification of simultaneous extracellular field recordings of preBötC activity and hypoglossal nucleus revealed a larger I/O ratio (*Figure 7E*, HO-2:CSE: 1.014±0.02, n=6, p=0.003) and near absence of transmission failures (*Figure 7F*, HO-2:CSE: 91.57 ± 3.20%, p=0.0007) when compared to activity in HO-2 null slices.

Similarly, in vivo L-PAG treatment, to acutely inhibit CSE activity, in HO-2 null mice improved transmission of preBötC activity to the hypoglossal nucleus in the rhythmic slice (*Figure 8A*, n=6) as indicated by larger cycle-to-cycle I/O ratio (*Figure 8B and L*-PAG = 1.01± 0.03, p=0.008) and greater transmission rates (*Figure 8C and L*-PAG 95.92 ± 2.18%, p<0.0001) when compared to the respective metrics from untreated HO-2 null mice. Intermittent transmission failure was also evident in patch clamp recordings from untreated HO-2 null hypoglossal neurons (*Figure 8D*, *left* shaded cycles) but not in HO-2 null hypoglossal neurons treated with L-PAG (*Figure 8D*, *right*). These reduced transmission events correlated with smaller individual inspiratory drive currents in HO-2 null hypoglossal neurons when compared to corresponding inspiratory synaptic drive currents from L-PAG-treated HO-2 mice (*Figure 8D–E*, HO-2: –36.71±2.14 pA vs. L-PAG –194.3±82.73 pA, p=0.0007). Together, these experiments implicated the involvement of CSE-dependent H$_2$S signaling with the effects of disrupted HO-2 /CO signaling affecting synaptic drive and output from hypoglossal motoneurons.

## Blockade of small conductance calcium-activated potassium channel (SK$_{Ca}$) activityrestores changes induced by HO-dysregulation in hypoglossal activity

As our experiments implicated the involvement of H$_2$S signaling, we next sought to determine how H$_2$S sensitive ion channels may contribute to impairing hypoglossal neuron activity caused by HO-dysregulation. H$_2$S has been shown to enhance activity of several different potassium channels, including SK$_{Ca}$ and ATP-sensitive potassium channel (K$_{ATP}$) activities *Mustafa et al., 2011*. As both SK$_{Ca}$ and K$_{ATP}$ are important in the regulation of hypoglossal neuron excitability *Haller et al., 2001*; *Lape and Nistri, 2000*, we examined how blocking these channels affected hypoglossal activity during HO-dysregulation. At the network level, administration of the selective SK$_{Ca}$ inhibitor, apamin (200 µM), increased the excitability of the hypoglossal neurons treated with ChrMP459. This enhanced activity exceeded the original baseline activity (i.e., prior to ChrMP459 administration) causing ectopic bursting in the

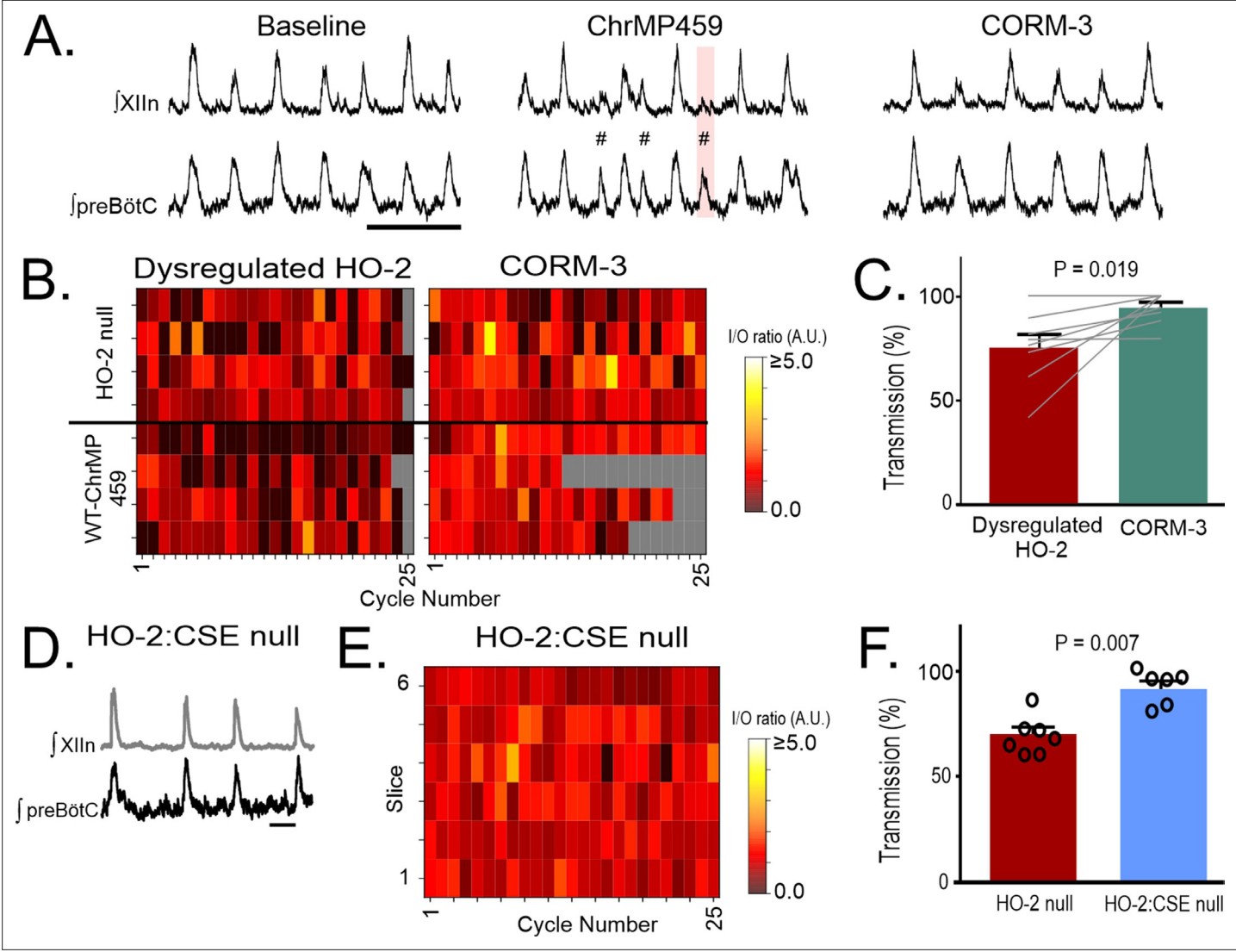

**Figure 7.** HO-dependent transmission failures can be recovered with CO-donor CORM-3 and are not present in HO-2:CSE null transmission. (**A**) Representative integrated traces from preBotC and XIIn in WT slices during Baseline (left), in ChrMP459 alone (middle) and in ChrMP459 +20 µM CORM-3 (CORM-3, right). Both subnetwork (#) and failed transmissions (pink rectangle) are highlighted. (**B**) Heat maps of cycle-to-cycle I/O ratios during dysregulated HO-2 (n=8: n=4 HO-2 null and n=4 WT-ChrMP459) before and after CORM-3 application. Gray boxes indicate non-events in recordings from slower rhythms where less than 25 events occurred during the analysis window. (**C**) Comparison of transmission from preBötC to XIIn from dysregulated HO-2 slices before (red) and after (green) bath application of CORM-3. (**D**) Representative integrated traces from preBötC and XIIn in slices from HO-2:CSE null; scale bar 2 sec. (**E**) Heat map of cycle-to-cycle I/O ratio from preBötC to XIIn in HO-2:CSE null. The I/O ratio from HO-2:CSE null is greater than I/O ratios from HO-2 null (n=7, p=0.003). Gray boxes indicate non-events in recordings from slower rhythms where less than 25 events occurred during the analysis window. (**F**) Comparison of transmission from preBötC to XIIn in HO-2 null (red, n=7, subset replotted from *Figure 2*) and HO-2:CSE null (light blue, n=6). HO-2 null data used for comparisons in E and F are a subset of the data originally shown in *Figure 2*. Statistical analysis for B and C via paired t-test, analysis for E and F via unpaired t-test; error bars: SEM; significance level P<0.05.

hypoglossal nucleus (*Figure 9—figure supplement 1A*) making analysis of population transmission and I/O ratios unreliable. Therefore, we proceeded to resolve the effects of apamin on the influence of ChrMP459 at the level of individual hypoglossal inspiratory motoneurons. While in some hypoglossal neurons exposed to ChrMP459, apamin substantially increased drive currents (>100 pA; *Figure 9A* **left**), in others, apamin modestly increased the drive current (<100 pA; *Figure 9A* **middle**). Despite this variability, apamin increased inspiratory drive currents received by ChrMP459 treated hypoglossal neurons (*Figure 9A*, **right,** n=6, ChrMP459: –85.77±38.54 pA vs. apamin: –219.97±97.76 pA, p=0.031). Apamin also enhanced the number of action potential generated per preBötC burst during ChrMP459 (*Figure 9B*, n=8, ChrMP-459: 12.57±3.68 action potential per burst vs. apamin 26.05±6.87

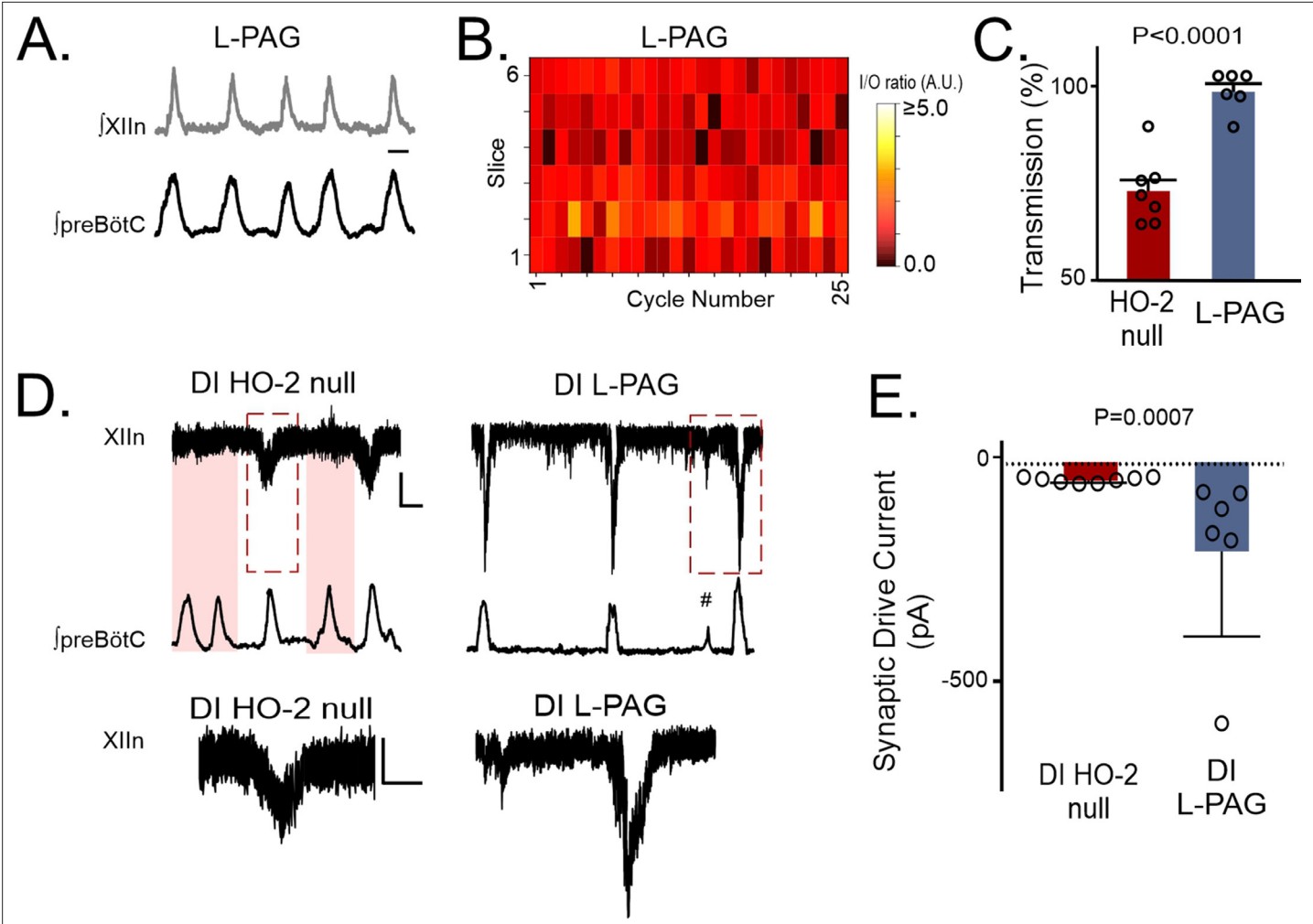

**Figure 8.** Transmission failures in HO-2 null mice are rescued with CSE inhibitor L-propargylglycine. (**A**) Representative integrated traces of preBötC (bottom) and XIIn (top) in slices from HO-2 null mice treated with L- propargylglycine (L-PAG, 30 mg/kg, n=5). Scale bar 2 sec. (**B**) Heat map of cycle-to-cycle I/O ratio in rhythmic slices from HO-2 null mice treated with L-PAG (n=6). (**C**) Comparison of transmission between HO-2 null (n=7, subset replotted from *Figure 2*; red) and L-PAG slices (n=6, blue). (**D**) Representative voltage clamp recordings of inspiratory drive currents received by hypoglossal neurons from DI HO-2 null (n=8, left) and DI L-PAG (n=6, right). Neurons were disinhibited from fast synaptic inhibition using 50 μM PTX and 1 μM Strychnine. Scale bar 100 ms x 20 pA. Skipped transmission between preBötC (bottom) the XIIn neuron (top) occurs in untreated DI HO-2 null (highlighted pink boxes) but not in neurons from DI L-PAG. Subnetwork transmission to XIIn neuron in DI L-PAG (#). Magnified representative (red dashed box) drive potentials from DI HO-2 null and DI L-PAG. scale bars: 100 msec x 10 pA. (**E**) Comparison of average synaptic drive currents received by XIIn motoneurons from DI HO-2 null mice (n=8, red) produce smaller drive potentials when compared to DI L-PAG (n=6, blue). Statistical analysis for all comparisons via unpaired t-test; error bars: SEM; significance level P<0.05.

The online version of this article includes the following figure supplement(s) for figure 8:

**Figure supplement 1.** Effects of genetic ablation of CSE and CSE pharmacological inhibition on the preBötC in the HO-2 null slice.

action potential per burst, p=0.016). This was consistent with the ability of apamin to reduce rheobase in ChrMP459-treated inspiratory hypoglossal neurons (*Figure 9C*; n=7, ChrMP459: 532.0±186.5 pA vs. Apamin: 307.09±79.62 pA, p=0.016).

To determine how blockade of $K_{ATP}$ impacted hypoglossal activity during ChrMP459, we used the $K_{ATP}$ channel blocker, tolbutamide (100 μM). Tolbutamide did not induce ectopic bursting in the hypoglossal nucleus during ChrMP459 (*Figure 9—figure supplement 1B*, n=5). Furthermore, tolbutamide (100 μM), did not improve the rate of transmission of preBötC activity (*Figure 9—figure supplement 1C*, *left*, ChrMP459: 69.24 ± 6.00% tolbutamide: 71.23 ± 6.86%, p=0.83) but did increase the I/O ratio (*Figure 9—figure supplement 1C*, *right*, ChrMP459: 0.687±0.05 tolbutamide: 0.870±0.05, p=0.037). Further, tolbutamide neither enhanced inspiratory drive currents in ChrMP459 (*Figure 9—figure*

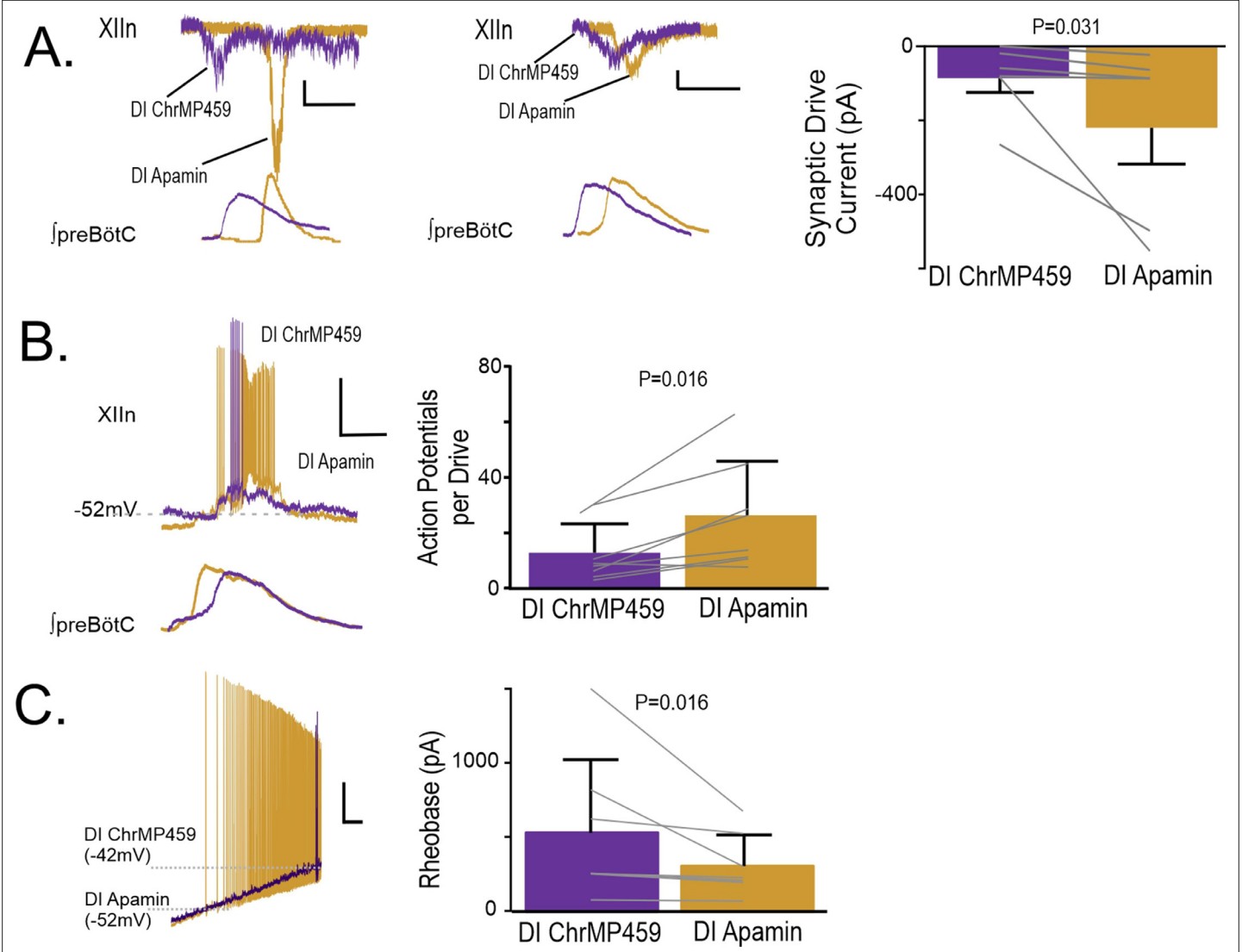

**Figure 9.** Apamin reverses changes to hypoglossal neurons' intrinsic and synaptic excitability caused by HO dysregulation. (**A**) Representative synaptic drive current received by hypoglossal neurons in DI ChrMP459 (purple) and in DI Apamin (200 μM, gold). Left, depicts an example of apamin increasing the drive current >100 pA. Scale bars: 1 s x 50 pA., whereas middle, representative inspiratory drive current hypoglossal neuron in DI ChrMP459 (purple) and in DI Apamin (gold) where apamin increased the drive current by less than <100 pA Scale bars: 1 s x 50 pA. Comparison of inspiratory drive currents from hypoglossal neurons exposed to DI ChrMP459 (purple) to DI Apamin (n=6, gold). The effect of ChrMP459 on baseline disinhibited drive current for each of these neurons were reported in *Figure 4A*. (**B**). (left) Representative current clamp recordings from a spontaneously active XIIn neuron with the preBötC network rhythm during DI ChrMP459 (purple) and in DI Apamin (n=8, gold). Scale bars: 20 mV x 500 ms. (right) Comparison of action potentials generated per preBötC burst during DI ChrMP459 (purple) and DI Apamin (n=8, gold). The effect of DI ChrMP459 on baseline action potential generated per preBötC burst for each of these neurons were reported in *Figure 4D* (**C**) (left) Representative traces of current clamp recordings in response to ramp current injection in DI (purple) and in DI Apamin (gold). Scale bar: 500 ms x 10 mV. (right) Rheobase comparison from inspiratory XIIn neurons during DI ChrMP459 (purple) and DI Apamin (n=7, gold). The effect of DI ChrMP459 on baseline rheobase for each of these neurons were reported in *Figure 4F*. Statistical analysis for all comparisons via paired t-test with Wilcoxon Correction; error bars: SEM; significance level P<0.05.

The online version of this article includes the following figure supplement(s) for figure 9:

**Figure supplement 1.** Apamin produce ectopic hypoglossal network activity during ChrMP459 and tolbutamide has limited effects on network and inspiratory hypoglossal neurons activity during ChrMP459.

*supplement 1D*, n=4, ChrMP459:–70.51±27.49 pA vs. tolbutamide: –83.06±36.29 pA, p=0.375) nor increased the number of action potentials per preBötC burst in ChrMP459 (*Figure 9—figure supplement 1E* n=4, ChrMP459: 7.063±2.08 action potential per burst vs. tolbutamide: 9.370±3.11 action potential per burst, p=0.125). Moreover, tolbutamide did not affect rheobase of inspiratory

**Table 2.** Comparison of ChrMP459 and HO-2 null effects on preBötC and XIIn transmission properties.

Properties of pharmacological effects of HO-2 dysregulation on preBötC and XIIn activity via ChrMP459 application were compared to genetic dysregulation in HO-2 null mouse experiments. Instantaneous frequency ($f_{inst}$), integrated burst amplitude, irregularity of integrated burst amplitude, and frequency of subnetwork bursting (% of total bursts) were analyzed for preBötC activity. I/O ratio of integrated preBötC area input to XIIn output and percent transmission were compared between the two regions. XIIn disinhibited neurons synaptic drive potentials were also assessed. Statistical analysis for all comparisons via unpaired t-test; values presented as mean ± SEM; significance level P<0.05.

| Metric | ChrMP459 (n) | HO-2 null (n) | p-value |
|---|---|---|---|
| $f_{inst}$* (Hz) | 0.24±0.019 (34) | 0.42±0.049 (11) | <0.0001 |
| Burst Amplitude* (mV) | 0.08±0.012 (34) | 0.05±0.010 (11) | 0.189 |
| IrS AMP† (A.U.) | 0.41±0.050 (34) | 0.39±0.070 (11) | 0.127 |
| Subnetwork‡ (%) | 19.43±3.373 (34) | 16.70±4.898 (11) | 0.670 |
| I/O Ratio§ (A.U.) | 0.59±0.064 (34) | 0.79±0.074 (11) | 0.096 |
| Transmission¶ (%) | 75.10±3.425 (34) | 64.11±6.002 (11) | 0.119 |
| Synaptic Drive Current** (pA) | −95.31±21.790 (19) | −36.71±2.141 (8) | 0.097 |

*Data originally reported in **Table 1**.
†data from **Figure 1C** and **Figure 2C**.
‡data from **Figure 5B** and **Figure 8—figure supplement 1**.
§data from **Figure 1D** and **Figure 9—figure supplement 1**.
¶data from **Figure 1E** and **Figure 9—figure supplement 1**.
**data from **Figure 4A** and **Figure 8E**.

hypoglossal neurons treated with ChrMP459 (*Figure 9—figure supplement 1F*, n=7; ChrMP459: 221.02±74.80 pA vs. tolbutamide: 180.40±68.63 pA, p=0.219). Thus, these results suggested that apamin could enhance activity of hypoglossal neurons during HO-inhibition; whereas, the efficacy of tolbutamide to impact activity during HO-inhibition was limited.

## Discussion

Our study reveals a previously uncharacterized neuromodulatory interaction between HO-2 and CSE-derived $H_2S$ regulating activity from the hypoglossal nucleus. HO-2 dysregulation promoted subnetwork activity and irregularities in rhythmogenesis from the preBötC while also disturbing excitatory synaptic drive currents and intrinsic excitability of inspiratory hypoglossal neurons. Our investigations indicate that these phenomena contribute to a reduction in the input-output relationship and an increased propensity for transmission failure between the preBötC and the hypoglossal motor nucleus. Blocking CSE activity mitigated many effects caused by HO-2 dysregulation; whereas, using an $H_2S$ donor mimicked the impairments to the input-output relationship and intermittent failures observed with HO-2 dysregulation. Together these findings demonstrate a role for centrally-derived interactions between HO-2 and CSE activity in regulating motoneuron output responsible for maintaining upper airway patency.

We used two approaches to dysregulate HO-2 activity in the rhythmic brainstem slice preparation. First, pharmacologically, by using the pan HO inhibitor, ChrMP459 in wild-type slices; and second, by performing experiments in slices from HO-2 null mice. Using these two approaches produced some empirical differences. The frequency of rhythmogenesis in the preBötC was faster and the hypoglossal IrS$_{AMP}$ was larger in HO-2 null networks when compared to wild-type networks. However, ChrMP459 did not impact these metrics. Despite these differences, both approaches produce similar effects on subnetwork preBötC activity, the I/O ratio, rate of transmission between preBötC and the hypoglossal nucleus and on synaptic drive currents in hypoglossal motoneurons (*Table 2*). Therefore, we considered these outcomes common to both approaches to represent the key effects caused by HO-2

dysregulation on inspiratory-related hypoglossal activity in the isolated brainstem slice and primarily focus on issues related to these phenomena throughout the remainder of the discussion.

Mice treated with intermittent hypoxia (IH) patterned after blood $O_2$ profiles associated with sleep apnea also show failed transmission and a reduced input-output relationship between the preBötC and hypoglossal nucleus *Garcia et al., 2016*. These IH-dependent effects also correlated to cycle to cycle irregularities in rhythmogenesis and caused subnetwork preBötC activity that failed to produce measurable output from the motor nucleus *Garcia et al., 2016*. Like that of IH, HO-2 dysregulation increases cycle to cycle irregularity of preBötC burst amplitude and produces subnetwork activity in the preBötC that often failed to produce measurable hypoglossal output. The failure to generate measurable output from the motor pool also occurred with larger preBötC bursts, albeit in a smaller percentage of occurrence (*Figure 1F* and *Figure 2F*) suggesting the impact of HO-2 dysregulation was not restricted to effects on the preBötC.

Simultaneous triple recordings from the preBötC, premotor field, and the hypoglossal nucleus demonstrated that in ChrMP459, premotor activity corresponding to the preBötC rhythm was reliable; whereas, hypoglossal activity often failed. These experiments suggested that HO-2 dysregulation produced deficits in synaptic physiology between intermediate premotor neurons and the hypo-glossal nucleus and possibly affected the postsynaptic excitability of hypoglossal motoneurons. Patch clamp recordings demonstrated that ChrMP459 reduced postsynaptic action potential generation and intrinsic excitability of inspiratory hypoglossal neurons. Reduced motoneuron excitability also corresponded with HO-2 dysregulation mediated reduction in synaptic drive currents in hypoglossal neurons. Although these experiments did not resolve the contribution of changes in presynaptic or postsynaptic properties to suppressing synaptic drive, the phenomenon appeared independent of effects related to inhibition as these experiments were performed in the presence of blockers for fast GABAergic and glycinergic receptors.

While HO-2 expression was not documented in either premotor neurons or the preBötC, HO-2 dysregulation impacted the respiratory network promoting subnetwork preBötC activity and increasing the irregularity of the rhythm. While the disinhibition experiments in ChrMP459 indicated that HO-2 dysregulation promotes a network state favoring synaptic inhibition to promote subnetwork activity (*Figure 5A*), disinhibition in HO-2 null slices neither improved the IrS$_{AMP}$ of the preBötC nor reduced the occurrence of subnetwork preBötC activity (*data not shown*). These divergent outcomes raise the possibility that long-term loss of HO-2 activity during development may have a broader impact on mechanisms of governing rhythmogenesis that cannot be corrected by acutely blocking synaptic inhibition in the brainstem network.

Based on prior observations demonstrating that lost HO-2 dependent CO activity enhances CSE-dependent H2S production and leads to respiratory disturbances *Peng et al., 2018*, it was predicted that the provision of CO or blockade CSE activity would improve subnetwork activity and amplitude irregularities of rhythmogenesis from the preBötC. While using, the CO donor, CORM-3 and blocking CSE activity in HO-2 null slices, on average, reduced both subnetwork activity and the IrS$_{AMP}$ in the preBötC, both phenomena were still statistically similar to that observed in preBötC rhythms recorded from unmanipulated HO-2 null slices (*Figure 8—figure supplement 1*). These findings suggests that HO-2 dysregulation may also involve actions that are independent from CO and CSE / H2S activities. In addition to CO, biliverdin-bilirubin and ferrous iron are bioactive molecules generated from activity of heme oxygenases *Snyder and Barañano, 2001*. Bilirubin acts as antioxidant protecting neurons from oxidative injury *Doré et al., 1999*, whereas ferrous iron can promote an oxidative state and cause injury *Gutteridge and Halliwell, 2018*. These molecules generated from HO-2 activity may be important to maintaining redox state and stable rhythmogenesis from the preBötC during develop-ment. Indeed, reactive oxygen species have specific and different actions on rhythmogenesis *Garcia et al., 2011* while the promote of a pro-oxidant state in the preBötC can lead to irregular rhythmo-genesis *Garcia et al., 2016*. Further work is needed to resolve the potential role for different bioactive molecules derived from HO-2 activity on neurophysiology of the preBötC.

Although we did not resolve potential differences in HO-2 expression among specific motoneu-rons that innervate different upper airway muscles, such as genioglossal neurons, divergent effects of ChrMP459 on non-inspiratory and inspiratory hypoglossal neuronal properties were observed. ChrMP459 decreased rheobase among non-inspiratory neurons, whereas in inspiratory hypoglossal neurons, it decreased the magnitude of drive currents, increased rheobase, and diminished the

number of action potentials generated during preBötC bursting. While these findings illustrate the potential for HO-dysregulation to differentially impact inspiratory and non-inspiratory hypoglossal neurons, these findings also emphasize a need to further resolve how HO-2 activity impacts different hypoglossal neurons innervating various muscle groups of the upper airway.

In the HO-2 null mouse, the incidence of OSA is absent with co-inhibition of CSE *Peng et al., 2018*, which is consistent with reports that CO generated by HO-2 inhibits CSE-dependent $H_2S$ production *Prabhakar, 2012*; *Morikawa et al., 2012*. After documenting CSE expression in hypoglossal neurons and demonstrating an increased abundance of $H_2S$ in the hypoglossal nucleus of HO-2 null mice, we demonstrated that a CO donor improves transmission and the input-output relationship between the preBötC and hypoglossal nucleus in HO-2 null slices (*Figure 7A–C*). Furthermore, using a $H_2S$ donor also increased transmission failures and reduced the I/O ratio similar to dysregulating HO-2 activity (*Figure 6D–E*). Endogenous $H_2S$ activity could originate from other $H_2S$ producing enzymes, such as cystathionone β-synthase (CBS) that is expressed primarily in astrocytes throughout the CNS *Enokido et al., 2005*; yet CBS inhibition appears to have limited impact on inspiratory activity from the hypoglossal nucleus *da Silva et al., 2017*. Additionally, our experiments manipulating CSE activity, by either treating HO-2 null mice with L-PAG or using HO-2:CSE null slices (*Figures 7D–E , and 8*), improved the I/O and reduced transmission failure from preBötC to the hypoglossal nucleus. Larger synaptic drive currents were also observed in HO-2 null hypoglossal neurons after treatment with L-PAG. Thus, in contrast to that in the preBötC, where HO-2 dysregulation may involve actions that independent from CSE / H2S, the mutual interaction between HO-2/CO and CSE/$H_2S$ appears to have a major role in regulating hypoglossal output by both modulating excitatory synaptic drive currents received by hypoglossal motoneurons and impacting their intrinsic excitability.

How might enhanced $H_2S$ signaling reduce excitatory synaptic currents and excitability of hypoglossal neurons? While it is possible that $H_2S$ may impact presynaptic release of glutamate from intermediate premotor neurons and/or postsynaptic receptor activity of hypoglossal neurons, the ChrMP459 mediated increase in rheobase among inspiratory hypoglossal neurons implicated the involvement of non-synaptic conductance(s) downstream of $H_2S$-based signaling caused by perturbations in HO-2 activity. $H_2S$ can enhance both $K_{ATP}$ and $SK_{Ca}$ activities *Mustafa et al., 2011*. In the hypoglossal neurons, $K_{ATP}$ is dynamically active causing periodic adjustment of neuronal excitability *Haller et al., 2001* while $SK_{Ca}$ also regulates excitability and firing properties of hypoglossal neurons *Lape and Nistri, 2000*. In ChrMP459, tolbutamide had a limited effect normalizing transmission as it improved the I/O ratio, but did not reduce transmission failure. Tolbutamide neither increased excitatory synaptic currents nor enhanced intrinsic excitability of hypoglossal neurons in ChrMP459. In contrast, apamin normalized synaptic drive currents and increased excitability of inspiratory hypoglossal neurons in ChrMP459. These results indicated that blockade of $K_{ATP}$ was limited in countering the effects on HO-2 dysregulation in the hypoglossal nucleus, whereas that blockade of $SK_{Ca}$ sufficiently mitigates many aspects of HO-2 dysregulation in hypoglossal neurons associated with preBötC activity.

In addition, the occurrence of obstructive apnea in HO-2 null mice *Peng et al., 2017*, spontaneous hypertensive rats exhibit an increased incidence of apneas and hypopneas that are associated with reduced CO levels due to a reduction in HO-2 activity and increased $H_2S$ generation *Peng et al., 2014*. While in both cases, these effects have been linked to interactions between CO and $H_2S$ in the peripheral nervous system, our findings indicate that central HO-2 dysregulation may also contribute to the incidence of apneas. Loss of HO-2 activity causes irregular rhythmogenesis from the preBötC while also reducing synaptic drive and intrinsic excitability in hypoglossal motoneurons. In the hypoglossal nucleus, the impact of this dysregulation appears to be largely normalized by blocking CSE activity suggesting that the interaction between HO-2 dependent CO production and CSE-dependent $H_2S$ activity have an important role in regulating hypoglossal activity. Additionally, while enhanced loop gain in HO-2 null mice has been attributed to increased chemo reflex sensitivity regulated by the carotid bodies *Peng et al., 2017*; *Peng et al., 2018*; *Osman et al., 2018*; *Prabhakar and Semenza, 2012*, this study implicates a concurrent target for HO-2 dysregulation in the central respiratory circuit. The combination of increased chemoreflex sensitivity, imbalanced preBötC excitation/inhibition activity, and reduced hypoglossal motoneuron excitability caused by disturbed HO-2 activity could all contribute to the increased loop gain that perpetuates transmission failures in respiratory motor output and disrupt upper airway patency in HO-2 null mice and OSA patients. Moreover,

this interaction may extend beyond apneas and may also be important for regulating the genioglossus and other muscle groups of the tongue during behaviors such as swallowing *Fregosi and Ludlow, 2014* and vocalization *Wei et al., 2022*, particularly when considering the potential for rapid signaling via CO and $H_2S$. Furthermore, should mutual interactions between these gasotransmitters and their respective enzymes exist in other motoneuron pools, our findings may be relevant to a variety of clinical conditions such as fentanyl-induced chest wall rigidity syndrome (i.e. wooden chest syndrome), amyotrophic lateral sclerosis, and spinal cord injury where upper airway control may be affected. Thus, while this study demonstrates the potential importance of central HO-2/CO and CSE/$H_2S$ interactions in regulating hypoglossal motoneurons, future work is needed to fully understand the role of gasotransmitters in the physiology of the hypoglossal nucleus and other motoneuron pools.

## Methods

### Study approval

In accordance with National Institutes of Health guidelines, all animal protocols were performed with the approval of the Institute of Animal Care and Use Committee at The University of Chicago (ACUP 72486, ACUP 71811).

### Experimental animals

Experiments were performed using neonatal (postnatal day 6 to postnatal day 12) wild-type mice (C57BL/6; Charles River), HO-2 null mice (from S. H. Snyder), and HO-2:CSE double-null mice. HO-2:CSE double-null mice were created by crossing HO-2 null females with CSE null males (from R. Wang, Department of Biology, Laurentian University, Sudbury, ON, Canada). Tissues from both sexes were used. No sex-based differences were observed; therefore, data from both sexes were pooled for analysis. All litters were housed with their dam in ALAAC-approved facilities on a 12 hr / 12 hr light-dark cycle.

### Pharmacological agents

Heme oxygenase activity was blocked using bath application of Chromium (III) Mesoporphyrin IX chloride (ChrMP459, 20 µM; Frontiers Sciences, Newark DE). A CO donor, CORM-3 (20 µM; Sigma-Aldrich St. Louis MO) was bath applied following ChrMP459 application. NaHS (10µM to 100 µM; Sigma-Aldrich), a $H_2S$ donor, was bath applied. In all patch clamp experiments, fast synaptic glycinergic and GABAergic inhibition was blocked by bath application of strychnine (1 µM; Sigma-Aldrich) and picrotoxin (50 µM; Sigma-Aldrich), respectively. Inhibition of CSE production was accomplished by in vivo L-propargylglycine L-PAG, 30 mg/kg (Sigma-Aldrich) administered (*i.p.* injection) 2.5–3 hrs prior to preparation of the rhythmic brainstem slice preparation. Inhibition of potassium channels $SK_{Ca}$ **and ATP**-sensitive potassium channel ($K_{ATP}$) was via bath application of Apamin (200 µM; Sigma-Aldrich) and Tolbutamide (100 µM; Sigma-Aldrich), respectively.

### Measurement of $H_2S$ production

Anaesthetized mice (urethane, 1.2 g•kg$^{-1}$ *i.p.*) were rapidly euthanized by decapitation. Following rapid removal of the brainstem, the tissue was flash frozen in liquid $N_2$. Flash frozen tissues were stored at –80 °C until coronal brainstem sections (300 µm thick) were cut with a cryostat at –20 °C and tissue punches of desired tissue were procured for immediate $H_2S$ measurements. The hypoglossal nucleus and control (inferior olive nucleus) brainstem tissue punches were made from the slices using a chilled micro-punch needle. Hypoglossal tissue from a single brainstem was not sufficient for effectively measuring $H_2S$ levels; therefore, we pooled bilateral micro punched tissue from two mice for each sample where $H_2S$ levels measured. $H_2S$ levels were determined as described previously *Yuan et al., 2016*. Briefly, cell homogenates from the pooled micro-punch tissue samples were prepared in 100 mM potassium phosphate buffer (pH 7.4). The enzyme reaction was carried out in sealed tubes. The assay mixture in a total volume of 500 µL contained (in final concentration): 100 mM potassium phosphate buffer (pH 7.4), 800 µM L-cysteine, 80 µM pyridoxal 5'-phosphate with or without L-PAG (20 µM) and cell homogenate (20 µg of protein), was incubated at 37 °C for 1 hr. At the end of the reaction, alkaline zinc acetate (1% mass / volume; 250 µL) and trichloroacetic acid (10% vol/vol) were sequentially added to trap $H_2S$ and stop the reaction, respectively. The zinc sulfide formed was

reacted with acidic N,N-dimethyl-p-phenylenediamine sulfate (20 µM) and ferric chloride (30 µM) and the absorbance was measured at 670 nm using Shimadzu UV-VIS Spectrophotometer. L-PAG inhibitable $H_2S$ concentration was calculated from a standard curve and values are expressed as nanomoles of $H_2S$ formed per hour per mg of protein.

## Immunohistochemistry

Anaesthetized mice (urethane, 1.2 g•kg$^{-1}$ *i.p.*) were perfused transcardially with heparinized phosphate-buffered saline (PBS) for 20 min followed by 4% paraformaldehyde in PBS. Brainstems were harvested, post-fixed in 4% paraformaldehyde overnight, and cryoprotected in 30% sucrose/PBS at 4 °C. Frozen tissues were serially sectioned at a thickness of 20 µm (coronal section) and stored at –80 °C. Sections were treated with 20% normal goat serum, 0.1% bovine serum albumin and 0.1% Triton X-100 in PBS for 30 min and incubated with primary antibodies against choline acetyltransferase (ChAT, 1:100; Millipore; #AB144P), and HO-2 (1:200, Novus Biologicals; # NBP1-52849) or CSE (1:250; gift from SH Snyder, Johns Hopkins University) followed by Texas Red-conjugated goat anti-mouse IgG (HO-2 and CSE) or FITC-conjugated goat anti-rabbit IgG (1:250; Molecular Probes, ChAT). After rinsing with PBS, sections were mounted in Vecta shield containing DAPI (Vector Labs) and analyzed using a fluorescent microscope (Eclipse E600; Nikon).

## Brainstem slices for electrophysiology

The isolated rhythmic brainstem slices were prepared as previously described *Garcia et al., 2017*. Briefly, anesthetized (1.5–3% isofluorane inhaled) animals were euthanized by rapid decapitation. Brainstems were dissected, isolated and placed into ice cold artificial cerebral spinal fluid (aCSF) (composition in mM: 118 NaCl, 25 NaHCO$_3$, 1 NaH$_2$PO$_4$, 1 MgCl$_2$, 3 KCl, 30 Glucose, 1.5 CaCl$_2$, pH = 7.4) equilibrated with 95% $O_2$, 5% $CO_2$. The isolated brainstem was glued to an agar block (dorsal face to agar) with the rostral face up and submerged in aCSF equilibrated with carbogen. Serial cuts were made through the brainstem until the appearance of anatomical landmarks such as the narrowing of the fourth ventricle and the hypoglossal axons. The preBötC and XIIn was retained in a single transverse brainstem slice (thickness: 560±40 µm). The slice was transferred into the recording chamber (~6 mL volume) where it was continuously superfused with recirculating aCSF (flow rate: 12–15 mL/min). Prior to recording, extracellular KCl was raised to 8 mM and the spontaneous rhythm was allowed to stabilize prior to the start of every experiment.

## Electrophysiology

Extracellular population activity was recorded with glass suction pipettes filled with aCSF. Pipettes were positioned over the ventral respiratory column containing the preBötC and over the ipsilateral medial dorsal column containing the hypoglossal nucleus. In some experiments, a third pipette was positioned between the preBötC and hypoglossal nucleus just lateral to the axon tract to record transmission through the premotor field containing intermediate premotor inspiratory neurons *Koizumi et al., 2013*; *Revill et al., 2015*. Extracellular population activity was recorded with glass suction pipettes filled with aCSF *Garcia et al., 2016*. The recorded signal was sampled at 5 kHz, amplified 10,000 X, with a lowpass filter of 10 kHz using an A-M instruments (A-M Systems, Sequim, WA, USA) extracellular amplifier. The signal was then rectified and integrated. Recordings were stored on a computer for *posthoc* analysis.

All intracellular recordings were made using the Multiclamp 700B (Molecular Devices). Acquisition and post hoc analyses were performed using the Axon pCLAMP10 software suite (Molecular Devices). Whole cell patch clamp recordings of hypoglossal motoneurons were obtained using the blind-patch technique with a sample frequency of 40 kHz. Recordings were made with unpolished patch electrodes, pulled from borosilicated glass pipettes with a capillary filament *Garcia et al., 2016*. The electrodes had a resistance of 3.5–8 MΩ when filled with the whole cell patch clamp pipette solution containing (in mM): 140 K-gluc acid, 1 CaCl$_2$, 10 EGTA, 2 MgCl$_2$, 4 Na$_2$-ATP, 10 HEPES. Neurons located at least two to three cell layers (about 75–250 µm) rostral from the caudal surface of the slice were recorded. The liquid junction potential was calculated to be –12 mV and was subtracted from the membrane potential. The series resistance was compensated and corrected throughout each experiment. In voltage clamp experiments, membrane potential was held at –60 mV. Current clamp experiments used a holding potential between 0 and –100 pA to establish the baseline resting membrane

potential between –55 and –70 mV. In some cases, we determined rheobases using a ramp protocol in our current clamp recordings. This ramp protocol consisted of a hyperpolarizing step (–100 pA) succeeded by the injection of a ramping depolarizing current (122 pA/sec; peak current 600 pA).

## Statistical analyses

Unless otherwise explicitly stated elsewhere, each n value represents an individual animal that served as a biological replicate for a given measurement. The irregularity score of amplitude ($IrS_{AMP}$) was calculated as described in *Garcia et al., 2016*. Transmission was expressed as a percentage of the hypoglossal network bursts corresponding to the total network bursts from either the preBötC or the premotor field. Bursts were considered corresponding if initial start time of bursts were within 500–750ms of each other (corresponding time was maximized until only one hypoglossal burst per preBötC was detected). Mean I/O and transmission values for each slice were calculated using a 120 s window. This analysis window was taken at the end of each baseline or pharmacological agent phase (each phase duration = 600 s). The input-output (I/O) ratio for each inspiratory event (defined by a network burst in preBötC) was calculated as the ratio of preBötC burst area to corresponding hypoglossal burst area as previously described *Garcia et al., 2016*. Calculation of the I/O ratio, was performed using the following equation:

$$IO_n = \int BA_{XIIn} / \int BA_{preBötCn}$$

where $IO_n$ is the I/O ratio of the $n^{th}$ cycle, $\int BA_{XIIn}$ is the integrated burst area in the hypoglossal nucleus of the $n^{th}$ cycle and preBötC the integrated burst area in the preBötC of the $n^{th}$ cycle. In cycles where preBötC did not correspond with hypoglossal output, $\int BA_{XIIn}$ was assigned a value of 0. Prior to the calculation of the I/O ratio, each $\int BA_{XII}$ was normalized to the mean hypoglossal integrated burst area of the analysis window, and each $\int BA_{preBötCn}$, was normalized to the mean preBötC integrated burst area of the analysis window. Heat maps were used to illustrate individual I/O ratios for up to 25 consecutive cycles in the analysis window for each experiment performed. To illustrate the cycle-to-cycle input-output relationships between networks, heat maps of I/O ratio values were plotted for each slice included in the experiment. Each row represents sequential cycles from a single slice experiment. As the rhythmic frequency across preparations varied, the number of events (i.e. cycle number) in the 120 s analysis window also varied; therefore, either the total number of cycles or up to 25 consecutive cycles from a given slice recording were plotted.

Statistics were performed using Origin 8 Pro (OriginLab, RRID: SCR_014212) or Prism 6 (GraphPad Software; RRID: SCR_015807). In cases where the distribution of data appeared normal, comparisons between two groups were conducted using either paired or unpaired two-tailed t-tests as appropriate. In cases, where the distribution of individual data points did not appear normal, the Wilcoxon match-paired signed rank test was performed. A one-way ANOVA was performed followed by *posthoc* Dunnett's test comparing experimental groups to control when a comparison of three or more groups. In plots where means are presented, the mean and error bars are presented, the error bars reflect the S.E.M. Differences were defined when the p-value was less than 0.05.

## Acknowledgements

This work was supported by NIH P01 HL144454 (NRP), NIH R01NS107421 (AJG), NIH R01HL163965 (AJG) and NIH R01DA057767 (AJG).

## Additional information

### Funding

| Funder | Grant reference number | Author |
| --- | --- | --- |
| National Heart, Lung, and Blood Institute | P01 HL144454 | Nanduri R Prabhakar |

| Funder | Grant reference number | Author |
|---|---|---|
| National Institute of Neurological Disorders and Stroke | R01NS107421 | Alfredo J Garcia III |
| National Institute on Drug Abuse | R01DA057767 | Alfredo J Garcia III |
| National Heart, Lung, and Blood Institute | R01HL163965 | Alfredo J Garcia III |

The funders had no role in study design, data collection and interpretation, or the decision to submit the work for publication.

## Author contributions

Brigitte M Browe, Formal analysis, Investigation, Methodology, Writing – original draft, Writing – review and editing; Ying-Jie Peng, Investigation, Methodology; Jayasri Nanduri, Investigation, Methodology, Writing – review and editing; Nanduri R Prabhakar, Conceptualization, Resources, Funding acquisition, Writing – review and editing; Alfredo J Garcia III, Conceptualization, Resources, Data curation, Formal analysis, Supervision, Funding acquisition, Investigation, Visualization, Methodology, Project administration, Writing – review and editing

## Author ORCIDs

Brigitte M Browe ⓘ http://orcid.org/0000-0003-1282-2290
Alfredo J Garcia III, ⓘ http://orcid.org/0000-0001-5620-7519

## Ethics

In accordance with National Institutes of Health guidelines, all animal protocols were performed with the approval of the Institute of Animal Care and Use Committee at The University of Chicago (ACUP 72486, ACUP 71811).

## Decision letter and Author response

Decision letter https://doi.org/10.7554/eLife.81978.sa1
Author response https://doi.org/10.7554/eLife.81978.sa2

## Additional files

### Supplementary files
• MDAR checklist

### Data availability

Numerical data used to generate figures is uploaded to Dryad. Source Data file names refer to current figure panels.

The following dataset was generated:

| Author(s) | Year | Dataset title | Dataset URL | Database and Identifier |
|---|---|---|---|---|
| Garcia AJ, Browe B | 2022 | Numerical Data from: Gasotransmitter Modulation of Hypoglossal Motoneuron Activity | https://dx.doi.org/10.5061/dryad.44j0zpchc | Dryad Digital Repository, 10.5061/dryad.44j0zpchc |

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
