## [Editor Report]

This is among the first papers to comprehensively describe a signaling pathway in motor neurons and the consequences of its deficiency. The finding of gaseous neurotransmitter controlling motor neuron function is novel and opens the door for investigations into other motor neuron pools.

---

## [Decision Letter]

**Decision letter after peer review:**

Thank you for submitting your article "Gasotransmitter modulation of Hypoglossal motoneuron activity" for consideration by *eLife*. Your article has been reviewed by 3 peer reviewers, one of whom is a member of our Board of Reviewing Editors, and the evaluation has been overseen by Jeannie Chin as the Senior Editor. The following individuals involved in review of your submission have agreed to reveal their identity: Ralph Fregosi (Reviewer #2); Silvia Pagliardini (Reviewer #3).

Essential revisions:

1) The evidence for transmission failure with ChrMP459 is somewhat weak (Figure 1E), though the data comparing WT and HO KOs is more convincing. Some discussion of these differences is warranted.

2) I am not sure how you can conclude that transmission uncoupling between premotor and XIIMNs is postsynaptic "These results suggested that dysregulated HO-2 modulates is a postsynaptic phenomenon in the hypoglossal nucleus." What if the gases also change excitatory and inhibitory synaptic inputs to hypoglossal Motoneurons (e.g., from GABAergic and/or glutamatergic neurons)?

3) Related to the above, the use of the term "intrinsic properties" in this context needs to be defined, otherwise, it may lead to confusion. Most consider intrinsic properties to be relatively stationary, at least on short timescales.

4) The rheobase data in Figure 4C is not convincing-did you run a stat test without the outliers?

5) Please indicate how tissue punches for H2S detection were collected in the methods- from fresh or fixed tissue? Following electrophysiological recordings or in naïve mouse tissue?

6) Figure.1 HO-2 immunostaining images do not have a particularly high resolution and appear blurry. Can the authors produce an image with a higher quality of staining?

7) Figures 1 and 3. Because the study is related to the pathway from the preBötC to the hypoglossal motoneurons, and previous work indicates the presence of central apneas in these mice, it would be helpful to determine the anatomical expression of HO-2 and CSE not only in the XII MN but also in the preBötC and XII premotoneurons. (If these data are not available, please consider discussing this as a limitation. Additional experiments are not required – MLB).

8)Figure 1B – failed transmission events are linked to events in PBC where peak amplitude appears to be reduced. Can the authors comment on this? Also, inspiratory output variability appears to be increased (and frequency reduced) following bath application of ChrMP. Were these parameters assessed?

9) Figure.1D. I find the heat maps a bit tricky to read. If it is an Input/Output ratio between PBC and motor nucleus for each respiratory-like event (baseline cycle number on the X axis), wouldn't you have an alternation between 0 and 1 (motor output happens or not)? Why then arbitrary units are indicated? Please clarify, maybe include more details on figure legends or methods section since this information is key for a better understanding of the results.

10) Figure.2 HO-2 null mice present also with a reduced respiratory frequency in addition to failed transmission. Was this quantified in medullary slices? Also, given the potentially wider effect of ChrMP pharmacological action (i.e., block of other enzymes), was the drug tested in HO-2 null mice?

11) Figure.6B, I may be wrong, but up to figure 5 all figures with the Heat maps demonstrate that disruption of CO production causes a decrease in the I/O ratios (figures 1D, 2B, 3C) with a reduction in transmission rate. Here, in Figure 6B with CORM3 you still have a decrease in the I/O ratios but an increase in transmission rate. Can you please clarify?

12) Figure. 7D1 Traces from PBC appear to be very different, can you comment on this?

Line 387. What about the data reported in Figure 3.C that include the connection with premotor neurons?

13) The contribution of the gas transmitters to our understanding of homeostatic plasticity would broaden the appeal of your work.

---

## [Author Response]

Essential revisions:1) The evidence for transmission failure with ChrMP459 is somewhat weak (Figure 1E), though the data comparing WT and HO KOs is more convincing. Some discussion of these differences is warranted.

In the original submission, the data from the Figure 1 (ChrMP459) and Figure 2 (HO-2 null) represented the initial experiments for each respective dataset documenting the effects of dysregulated HO-2 activity via pharmacological or genetic means. In many of the later experiments throughout the study, the same protocols were used but the effects of either ChrMP459 or the HO-2 null slices on the parameters were not reported in the original figure (i.e., Figures 1 & 2).

In this revision, we have now aggregated results from all experiments across the study for ChrMP459 and the HO-2 null slices and report their respective effects on the preBötC rhythm; extracellular hypoglossal activity; transmission failure; and the input-output relationship between preBötC and hypoglossal nucleus. In HO-2 null slices, the instantaneous frequency of rhythmogenesis in the preBötC was faster, and the irregularity of burst amplitude (IrS_AMP_) in the hypoglossal nucleus larger compared to the respective metrics from unperturbed wild-type slices (see: Figure 2 and Table 1); whereas, these metrics were unaffected in the ChrMP459 experiments (see Figure 1 and Table 1).

Given the Reviewer’s comment, we performed a meta-analysis comparing the findings from ChrMP459 and HO-2 null experiments (See Table 2). At the level of preBötC, the occurrence of subnetwork activity, burst amplitude and IrS_AMP_ were similar between ChrMP459 and HO-2 null experiments. The I/O ratio and rate of transmission failure were also similar between approaches. Finally, the size of inspiratory synaptic drive currents in hypoglossal neurons were not difference between ChrMP459 and HO‑2. Thus, while HO-2 null slices showed faster rhythms and more irregularity at the level of hypoglossal nucleus, this meta-analysis demonstrates that both approaches to dysregulate HO-2 activity produced similar and consistent outcomes at the network and individual neuron levels.

2) I am not sure how you can conclude that transmission uncoupling between premotor and XIIMNs is postsynaptic "These results suggested that dysregulated HO-2 modulates is a postsynaptic phenomenon in the hypoglossal nucleus." What if the gases also change excitatory and inhibitory synaptic inputs to hypoglossal Motoneurons (e.g., from GABAergic and/or glutamatergic neurons)?

We agree with the reviewer’s comment and have removed the original statement and replaced it with a more in-depth analysis and discussion of factors contributing dysregulation of the hypoglossal nucleus, including HO-2 modulation of the preBötC.

3) Related to the above, the use of the term "intrinsic properties" in this context needs to be defined, otherwise, it may lead to confusion. Most consider intrinsic properties to be relatively stationary, at least on short timescales.

Thank you for bringing this to our attention. We have revised and now use the term intrinsic excitability.

4) The rheobase data in Figure 4C is not convincing-did you run a stat test without the outliers?

While there are two data points which appear larger than much of the baseline rheobase data, they were included in the data analysis as these baseline rheobase values did not exceed two standard deviations from the mean. Performing statistical analyses without these outliers still reveal a significant difference in rheobase before and after ChrMP459 application (P=0.01, Baseline: 123.2 ± 20.32 pA; ChrMP459: 254.0 ± 54.35 pA).

5) Please indicate how tissue punches for H2S detection were collected in the methods- from fresh or fixed tissue? Following electrophysiological recordings or in naïve mouse tissue?

Tissue punches for H_2_S detection were collected from naive flash-frozen tissue (i.e., not fixed or fresh). We have revised the Results section and methods to make this clear.

“Anaesthetized mice (urethane, 1.2g•kg^−1^ *i.p.*) were rapidly euthanized by decapitation. Following rapid removal of the brainstem, the tissue was flash frozen in liquid N_2_. Flash frozen tissues were stored at -80^o^C until coronal brainstem sections (300μm thick) were cut with a cryostat at –20°C and tissue punches of desired tissue were procured for immediate H_2_S measurements. The hypoglossal nucleus and control (inferior olive nucleus) brainstem tissue punches were made from the slices using a chilled micro-punch needle. Hypoglossal tissue from a single brainstem was not sufficient for effectively measuring H_2_S levels; therefore, we pooled bilateral micro punched tissue from two mice for each sample where H_2_S levels measured.”

6) Figure.1 HO-2 immunostaining images do not have a particularly high resolution and appear blurry. Can the authors produce an image with a higher quality of staining?

We believe this issue was related a file conversion problem. We have rectified this issue.

7) Figures 1 and 3. Because the study is related to the pathway from the preBötC to the hypoglossal motoneurons, and previous work indicates the presence of central apneas in these mice, it would be helpful to determine the anatomical expression of HO-2 and CSE not only in the XII MN but also in the preBötC and XII premotoneurons. (If these data are not available, please consider discussing this as a limitation. Additional experiments are not required – MLB).

Unfortunately, immunohistological expression of neither CSE nor HO-2 were examined at the level of the preBötC. As the recommended by the reviewer, we now recognize this limitation in our discussion.

8) Figure 1B – failed transmission events are linked to events in PBC where peak amplitude appears to be reduced. Can the authors comment on this? Also, inspiratory output variability appears to be increased (and frequency reduced) following bath application of ChrMP. Were these parameters assessed?

Our revision now includes an extensive analysis related to the preBötC (see Figures 1 and 2, and Table 1 and Figure 8—figure supplement 2:). Both ChrMP459 and HO-2 null experiments increased irregularity score of burst amplitude in the preBötC. This was associated with an increased the occurrence of subnetwork preBötC activity (defined as network bursts having ≤50% burst area). Many failed transmission events coincided with subnetwork preBötC activity; however, transmission failures were observed across the continuum burst areas in the preBötC in both ChrMP459 and HO-2 null experiments.

9) Figure.1D. I find the heat maps a bit tricky to read. If it is an Input/Output ratio between PBC and motor nucleus for each respiratory-like event (baseline cycle number on the X axis), wouldn't you have an alternation between 0 and 1 (motor output happens or not)? Why then arbitrary units are indicated? Please clarify, maybe include more details on figure legends or methods section since this information is key for a better understanding of the results.

The heat maps illustrating the cycle-to-cycle variability of the input-output relationship for up to 25 consecutive cycles in the two-minute analysis window. The x-axis reflects cycle number while the y-axis represents an individual biological replicate of the slice experiment. The I/O ratio is a non‑binary number value reflecting the relative strength for preBötC activity to that of corresponding hypoglossal activity. We have now included the expanded description (provided below) in the methods section to improve clarity for the reader.

“The input-output (I/O) ratio for each inspiratory event (defined by a network burst in preBötC) was calculated as the ratio of preBötC burst area to corresponding hypoglossal burst area as previously described^28^. Calculation of the I/O ratio, was performed using the following equation:

IO*_n_* = ∫BA_XII*n*_ / ∫BA_preBötC*n*_

where IO_n_ is the I/O ratio of the n^th^ cycle, ∫BA_XII*n*_ is the integrated burst area in the hypoglossal nucleus of the n^th^ cycle and ∫BA_preBötC*n*_ the integrated burst area in the preBötC of the n^th^ cycle. In cycles where preBötC did not correspond with hypoglossal output, ∫BA_XII*n*_ was assigned a value of 0. Prior to the calculation of the I/O ratio, each ∫BA_XII_ was normalized to the mean hypoglossal integrated burst area of the analysis window, and each ∫BA_preBötC_, was normalized to the mean preBötC integrated burst area of the analysis window.”

As illustrated by the equation provided, I/O ratio is a unitless value. As factors such as electrode positioning and recording tip size, can impact integrated burst area from, the normalization steps prior to calculation allow for unbiased comparison across individual slice experiments.

10) Figure.2 HO-2 null mice present also with a reduced respiratory frequency in addition to failed transmission. Was this quantified in medullary slices? Also, given the potentially wider effect of ChrMP pharmacological action (i.e., block of other enzymes), was the drug tested in HO-2 null mice?

The original representative trace from Figure 2 did not accurately reflect the mean frequency of rhythmogenesis in HO-2 null mice. Our analyses (see: Table 1) indicate that HO2-null preBötC rhythms are faster than corresponding wild‑type rhythms. We have now replaced the representative traces to more accurately reflect this.

11) Figure.6B, I may be wrong, but up to figure 5 all figures with the Heat maps demonstrate that disruption of CO production causes a decrease in the I/O ratios (figures 1D, 2B, 3C) with a reduction in transmission rate. Here, in Figure 6B with CORM3 you still have a decrease in the I/O ratios but an increase in transmission rate. Can you please clarify?

Using the pharmacological CO donor CORM-3 increased the I/O ratio when applied to preparations where HO-2 was dysregulated the data illustrates this, please see line 352.

“Dysregulated HO-2 activity, caused by either pharmacological (ChrMP459) or genetic (HO-2 null mice) manipulation, is improved by CORM-3 as indicated by larger I/O ratios (Figure 6B, n=8; dysregulated HO-2: 0.70 ± 0.09 vs. CORM-3: 1.05 ± 0.07, P=0.01) and improved transmission (Figure 6C, dysregulated HO-2: 75.05 ± 6.3% vs. CORM-3: 94.21 ± 2.67%, P=0.02).”

Upon inspection, we found that the I/O heat map images for Dysregulated HO-2 and the CORM-3 rescue were flipped, this has now been rectified and the proper images conveying this data are in place.

12) Figure. 7D1 Traces from PBC appear to be very different, can you comment on this?Line 387. What about the data reported in Figure 3.C that include the connection with premotor neurons?

The screen captures of representative traces in Figure 7D1 and Figure 3C were obtained using different parameters which caused the appearance of differences between in preBötC traces. We revised these figures to better show the resulting comparisons.

13) The contribution of the gas transmitters to our understanding of homeostatic plasticity would broaden the appeal of your work.

We thank the reviewer for bringing this to our attention and agree with the sentiment for broadening the appeal of our study. However, as our study does not directly test aspects of homeostatic plasticity (i.e., the capacity of neurons to regula), we felt that including too much discussion of this section such discussion may be overreaching, but we have recrafted many aspects of the manuscript in an attempt to broadening the importance of our work.